# The DIAPH3 linker specifies a β-actin network that maintains RhoA and Myosin-II at the cytokinetic furrow

Riya Shah[1,4], Thomas C. Panagiotou [2,4], Gregory B. Cole[1], Trevor F. Moraes [1], Brigitte D. Lavoie[2], Christopher A. McCulloch [3] & Andrew Wilde [1,2] ✉

Cytokinesis is the final step of the cell division cycle that leads to the formation of two new cells. Successful cytokinesis requires significant remodelling of the plasma membrane by spatially distinct β- and γ-actin networks. These networks are generated by the formin family of actin nucleators, DIAPH3 and DIAPH1 respectively. Here we show that β- and γ-actin perform specialized and non-redundant roles in cytokinesis and cannot substitute for one another. Expression of hybrid DIAPH1 and DIAPH3 proteins with altered actin isoform specificity relocalized cytokinetic actin isoform networks within the cell, causing cytokinetic failure. Consistent with this we show that β-actin networks, but not γ-actin networks, are required for the maintenance of non-muscle myosin II and RhoA at the cytokinetic furrow. These data suggest that independent and spatially distinct actin isoform networks form scaffolds of unique interactors that facilitate localized biochemical activities to ensure successful cell division.

The actin cytoskeleton derives from the polymerization of actin monomers into filaments and their subsequent organization into a diverse array of structures. Actin networks engage in different cellular processes that include, but are not limited to, endocytosis, cell migration, and cytokinesis[1]. In vivo, actin monomers require nucleators to stimulate filament formation. Different nucleators are regulated by divergent molecular pathways to generate localized actin structures with specific cellular functions. The formin family of actin nucleators and the tandem monomer binding proteins stimulate the formation of linear filaments[2]. In contrast, the ARP2/3 complex stimulates the formation of branched actin filament networks[3,4]. The organizational state of these actin networks is further differentiated by the recruitment of different actin-binding proteins to actin filaments[5].

The diversity of the actin cytoskeleton is not, however, limited to the physical organization of the filaments, it extends to the utilization of different actin isoform monomers. There are seven actin genes in vertebrates: four have restricted tissue expression (α-skeletal muscle, α-smooth muscle, α-cardiac muscle, and γ-smooth muscle, that are

94% identical to β-actin), while the non-muscle β-actin and γ-actin genes (99% identical) are ubiquitously expressed[6]. A seventh actin gene, *actbl2* that is 92% identical to β-actin is poorly understood[7]. The different actin isoforms generate distinct networks[8] that have been attributed to different functions[6,9–12]. One instance occurs during cell division, where β-actin and γ-actin display different localization patterns and perform unique functions[8,13–16]. These observations are yet more intriguing given that β-actin and γ-actin only differ by four very conserved amino acids clustered in the first ten amino acids of the protein: aspartate$_{2-5}$ and isoleucine$_{10}$ in β-actin are replaced by glutamate$_{2-5}$ and valine$_{10}$ in γ-actin. An isoform-specific function has also been demonstrated in other biological contexts. The depletion of individual actin isoforms in tissue culture cells and/or injection of isoform-specific N-terminal peptides have been used to demonstrate non-redundant roles for β- and γ-actin in the maintenance of apical vs tight junctions of epithelial cells[17], in melanoma cell motility[18], during meiosis[9], and also in cell contractility[19,20]. In animal models, knocking out the β- and γ-actin genes independently caused different

[1]Department of Biochemistry, University of Toronto, 661 University Ave, Toronto, ON M5G 1M1, Canada. [2]Department of Molecular Genetics, University of Toronto, 661 University Ave, Toronto, ON M5G 1M1, Canada. [3]Faculty of Dentistry, University of Toronto, Toronto, ON M5G 1G6, Canada. [4]These authors contributed equally: Riya Shah, Thomas C. Panagiotou. ✉e-mail: andrew.wilde@utoronto.ca

phenotypes: β-actin knock out mice are embryonic lethal[21–24], whereas γ-actin knockout mice are viable, albeit subject to increased mortality and progressive hearing loss[21,25]. Interestingly, converting the mouse β-actin gene to express γ-actin protein resulted in viable mice with only modest phenotypes[26]. A further study converted the mouse β-actin gene to express γ-actin protein in animals where the γ-actin gene was also ablated and found this rescued some phenotypes associated with γ-actin loss, but introduced further phenotypes[27]. Therefore, the degree to which one actin isoform can substitute for the other remains unclear.

Linear actin filaments can be generated by formins, a family of 15 proteins in mammals to which diverse cellular functions have been attributed. The formin family is defined by the presence of a formin homology 2 (FH2) domain that confers the actin polymerization activity of the protein[28,29]. The activity of the Diaphanous sub-family of formins is tightly regulated through an autoinhibition mechanism mediated through intramolecular interactions between the diaphanous autoinhibitory domain (DAD) and the DAD interacting domain (DID). Activation is achieved through the sequential binding of RhoA, a small G-protein, followed by an enhancer protein that together releases the DID-DAD interaction, allowing actin monomer access to the FH2 domain and promoting actin polymerization[13,30,31].

During cytokinesis in tissue culture cells, the formins DIAPH1 and DIAPH3 perform vital roles. Interestingly, whilst *DIAPH3*⁻/⁻ mice die *in utero*, *DIAPH1*⁻/⁻ reaches adulthood, but with some defects in the brain[32,33]. During cytokinesis in tissue culture cells DIAPH1 localizes to the cell cortex to generate a γ-actin network whilst DIAPH3 localizes to the cytokinetic furrow where it generates a β-actin network[13,14]. The coordinated action of both actin isoform networks remodels the plasma membrane to drive the generation of two new cells[14]. During cell division, the regulation of β- and γ-actin networks are governed by different pathways. At the cytokinetic furrow, β-actin filaments are generated by the DIAPH3 formin that is activated by the binding of RhoA-GTP and anillin[13]. In contrast, the simultaneous disassembly of the polar γ-actin network is driven by the inactivation of DIAPH1 through the delivery of CLIP170 to the polar cortex, where it disrupts the activating IQGAP1-DIAPH1 interaction[14]. Our work and that of others support a model where distinct actin isoform networks are generated through the localized regulation of independent actin nucleators. Consistent with this idea, we have previously shown that DIAPH3 generates β-actin but not γ-actin homopolymers both in vitro and in cellula[13]. These observations suggest that the distinct β- and γ-actin isoform networks generated during cytokinesis are established by both the independent regulation as well as the intrinsic isoform selectivity of the different formins. Whether one actin isoform can substitute for another during cytokinesis and potentially other cellular processes remains an open question.

Key to understanding the function of specific actin isoform networks across systems is the determination of when, where, and how these networks are generated within the cell. To begin to answer this question, we first defined the specificity elements within DIAPH1 and DIAPH3 that underlie their actin isoform preferences, then generated tools to relocate β- and γ-actin networks within a dividing cell. We show that β- and γ-actin networks cannot substitute for one another during cell division, and that the β-actin network at the cytokinetic furrow is specifically required for the maintenance of RhoA and non-muscle myosin II at that site. These findings are suggestive of a model where distinct actin isoform networks serve to spatiotemporally regulate biochemical activities to execute diverse cellular processes.

## Results

### The DIAPH3 FH2 domain confers β-actin isoform specificity
We previously demonstrated in cellula that DIAPH3 preferentially generates β-actin filaments at the cytokinetic furrow, whereas DIAPH1 generates cortical γ-actin filaments during cell division[13,14]. As the FH2

domain is responsible for the actin filament nucleation and polymerization activity of formins[34], we hypothesized that the FH2 domain also dictated actin isoform specificity for each formin. To test this model, we exploited an assay whereby formin activity is targeted to the surface of mitochondria by fusing the formin to the mitochondrial targeting domain of TOM20[13]. Upon expression of the TOM20-GFP fusion protein in HeLa cells, no significant colocalization of GFP-labeled mitochondria and actin isoform immunofluorescence signal was observed. Therefore, little actin is associated with mitochondria in control cells (Fig. 1b, c). In contrast, in HeLa cells expressing a constitutively active C-terminal fragment of DIAPH3 (residues 540 to 1171, Fig. 1a) fused to the mitochondrial surface targeting domain of TOM20 and GFP, β-actin filament formation was observed on the surface of mitochondria (Manders' overlap coefficient (MOC) = 0.78 ± 0.15 compared to 0.06 ± 0.08 for empty vector, $p = 1.45 \times 10^{-11}$), but γ-actin filaments were observed to a much lesser degree (MOC = 0.25 ± 0.15 compared to 0.11 ± 0.12 for empty vector, $p = 6.68 \times 10^{-4}$; Fig. 1c, d), confirming that DIAPH3 preferentially generates β-actin filaments. In HeLa cells expressing TOM20-GFP fused to constitutively active DIAPH1, β- and γ-actin filaments were equally observed on mitochondria (β-actin colocalization MOC = 0.88 ± 0.13 compared to 0.06 ± 0.08 for empty vector control, $p = 1.45 \times 10^{-11}$; γ-actin colocalization MOC = 0.82 ± 0.15 compared to 0.11 ± 0.12 for empty vector control, $p = 1.45 \times 10^{-11}$; Fig. 1c, e).

To determine if the FH2 domain is sufficient to dictate the specificity for the β-actin isoform, we made constitutively active formin chimeras by swapping the FH2 domains between DIAPH1 and 3. Upon expression of the DIAPH1 chimera containing the DIAPH3 FH2 domain (DIAPH1-3F), β-actin filaments were observed on mitochondria (MOC = 0.67 ± 0.18) while γ-actin filaments were appreciably reduced (MOC = 0.25 ± 0.15 compared to 0.82 ± 0.15 for wild-type DIAPH1, $p = 7.25 \times 10^{-11}$; Fig. 1 c, e), indicating that the FH2 domain of DIAPH3 restricts the actin nucleation specificity of the formin to the β-actin isoform. In contrast, upon expression of the DIAPH3 chimera containing the DIAPH1 FH2 domain (DIAPH3-1F), both β and γ-actin filaments localized to mitochondria (β-actin colocalization: MOC = 0.69 ± 0.16 compared to 0.06 ± 0.08 for empty vector, $p = 2.90 \times 10^{-11}$; γ-actin colocalization MOC 0.82 ± 0.11 compared to 0.11 ± 0.12 for empty vector control, $p = 1.45 \times 10^{-11}$; Fig. 1c, d). These data suggest that the FH2 domain of DIAPH3 preferentially generates β-actin filaments, while the FH2 domain of DIAPH1 promotes both β- and γ-actin filament formation.

To determine if the DIAPH3 FH2 domain is sufficient to specify β-actin filament generation in the absence of cellular factors, we performed in vitro actin pelleting assays. We purified GST-tagged, constitutively active DIAPH1 and DIAPH3 formins, as well as the DIAPH1-3F and DIAPH3-1F chimeras, and incubated each of them with either actin purified from human platelets which is enriched for β-actin (~83% β-actin and ~17% γ-actin) or from chicken gizzards which is enriched for γ-actin (~24% β-actin and ~76% γ-actin). Polymerized actin was harvested by ultracentrifugation (pellet fraction, P) and the extent of polymerized to unpolymerized actin in the supernatant (S) compared by SDS-PAGE. All recombinant fragments were able to generate actin polymer (Supp Fig. 1a, b).

To directly determine the extent of assembly of each actin isoform across the different conditions, the actin polymerization assays were re-assessed by Western blotting using actin isoform-specific antibodies[8]. While DIAPH3 predominantly generated β-actin filaments, DIAPH1 generated both β- and γ-actin filaments (Fig. 2a). To determine if the DIAPH3 FH2 domain is sufficient to distinguish between actin isoforms, recombinant, constitutively active DIAPH1 and 3 chimeras fused to GST were purified and incubated with the different actin sources. While all formin chimeras were functional, as judged by their ability to generate actin polymer (Supp Fig. 1a, b), DIAPH3-1F generated both β- and γ-actin polymer (Fig. 2b), suggesting that the

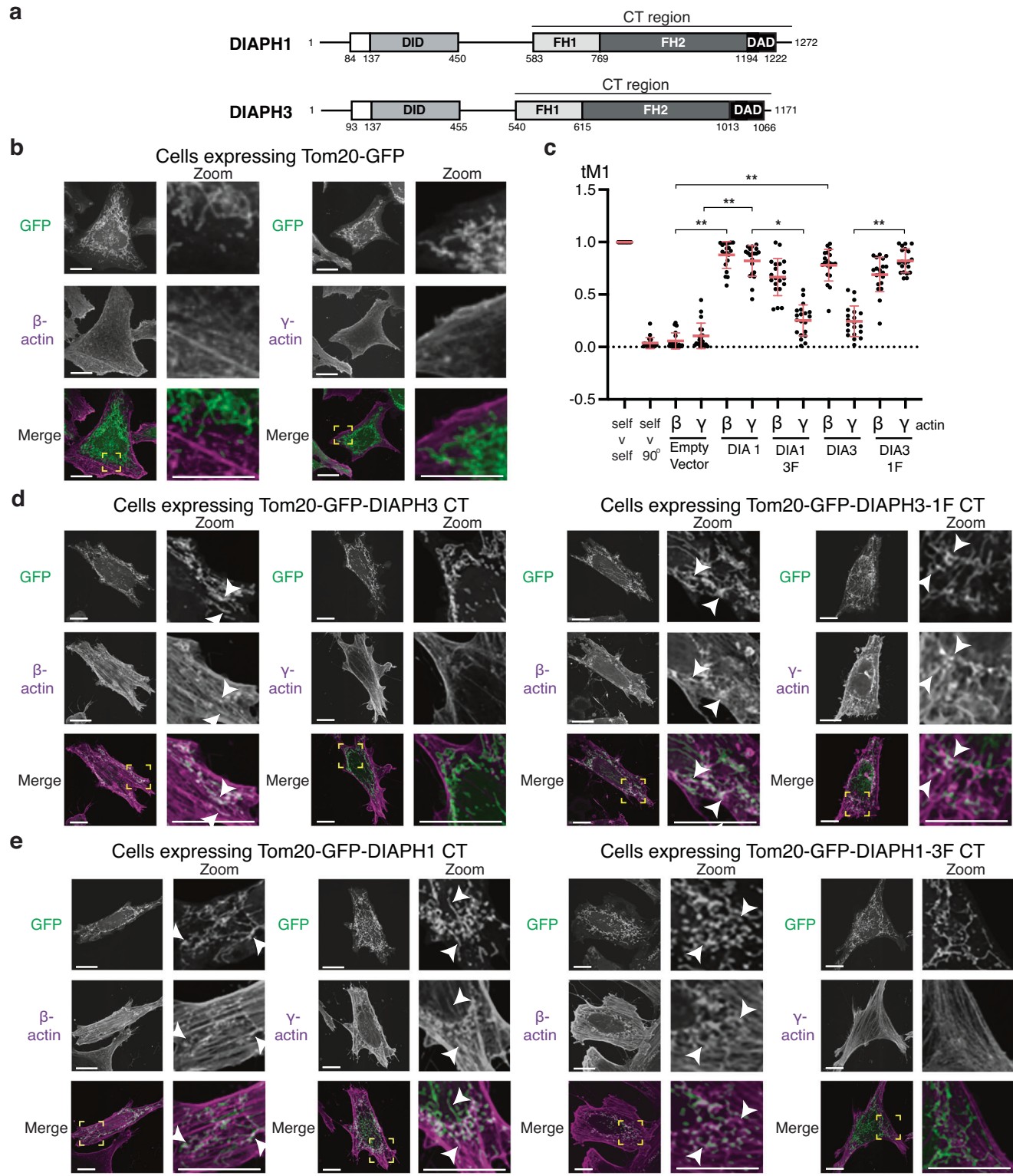

replacement of the DIAPH3 FH2 with that of DIAPH1 expands the iso-form specificity of DIAPH3 to that observed in the WT DIAPH1 formin. In contrast, while WT DIAPH1 generates both β- and γ-actin polymers (Fig. 2a, c), the replacement of its FH2 domain with that of DIAPH3 (DIAPH1-3F) restricted the substrate specificity of the formin such that only β-actin polymers were generated (Fig. 2c).

We next assessed whether the actin filaments produced by the chimeras are comprised of one or both actin isoforms (homo- versus heteropolymers) by determining the composition of the actin

filaments through immunofluorescence using actin isoform-specific antibodies. Interestingly, all formins generated homopolymers: DIAPH1 generated both β- and γ-actin homopolymers, whereas DIAPH3 generated only β-actin homopolymers (Supp Fig. 1c–e). Consistent with this, the chimeric formins generated homopolymers with actin isoform specificity following the FH2 domain; DIAPH1-3F preferentially generated β-actin homopolymers, whereas DIAPH3-1F generated both β- and γ-actin polymers (Supp Fig. 1c–e). Combined, these data suggest that the isoform specificity of formins derives predominantly from

**Fig. 1 | The FH2 domains of DIAPH1 and 3 directs their actin isoform specificity.**
**a** Schematics of the domain organization of DIAPH1 and 3. The domain boundaries are denoted as the amino acid number in the sequence. DAD diaphanous auto-inhibitory domain, DID-DAD interacting domain, FH1 formin homology domain 1, FH2 formin homology domain 2, and CT carboxy-terminal domain that is used in subsequent experiments (amino acids 583–1272 and 540–1171 for DIAPH1 and 3 respectively). **b** Micrographs of HeLa cells expressing GFP targeted to the surface of mitochondria by a fragment of Tom20 (green), fixed and stained for β- or γ-actin (magenta). Scale bars represent 10 μm. The presented micrographs are representative of three independent experiments. **c** Relative colocalization, determined by a Manders' overlap coefficient (tM1) of β- and γ-actin to mitochondria in HeLa cells expressing Tom20-GFP fused to the different CT domains of wildtype DIAPH1 (DIA1), wildtype DIAPH3 (DIA3), DIAPH1 with the DIAPH3 FH2 domain (DIA1 3F) and

DIAPH3 with the DIAPH1 FH2 domain (DIA3 1F). Data were presented as mean (solid bar) ± SD (error bars). $n = 20$ cells analyzed per condition across three independent experiments, $*p = 7.25 \times 10^{-11}$, $**p = 1.45 \times 10^{-11}$ as calculated by two-sided Mann–Whitney non-parametric tests. **d** Micrographs of HeLa cells expressing either a Tom20-GFP-DIAPH3-CT fusion protein or a Tom20-GFP-DIAPH3-1F CT (green), fixed and stained for β- or γ-actin (magenta). Scale bars represent 10 μm. The presented micrographs are representative of three independent experiments. **e** Micrographs of HeLa cells expressing either a Tom20-GFP-DIAPH1-CT fusion protein or a Tom20-GFP-DIAPH1-3F CT (green), fixed and stained for β- or γ-actin (magenta). Scale bars represent 10 μm. White arrowheads point to regions of actin colocalizing to mitochondria. The region boxed is yellow and is magnified in the zoom panel. The presented micrographs are representative of three independent experiments.

their FH2 domains, and that the DIAPH3 FH2 domain has the intrinsic ability to distinguish between the β- and γ-actin isoforms to pre-ferentially generate β-actin filaments.

## The DIAPH3 FH2 linker region confers β-actin isoform specificity
The FH2 domain can be broken down into a series of sub-regions, each with defined functions[35]. To determine which of these regions is the sensor for different actin isoforms, we assessed which region is closest to the extreme actin monomer N-terminus, the only region of sequence divergence between β- and γ-actin. As the extreme N-termini of actin monomers are not clearly resolved in crystal structures when in complex with a formin[36,37], we extrapolated the most likely positions of the actin N-terminus and the FH2 domain using the co-crystal structure of α-skeletal muscle actin and the formin FMNL3[37]. This positioned the N-terminus of the actin monomer closest to the flexible linker region of the FH2 domain, which is also poorly resolved in the crystal structures (Fig. 3a, b). Supporting this notion, an in silico molecular dynamics study predicted contacts between the actin N-termini and the linker region of formins, including DIAPH1[38]. To further confirm that this model may apply to DIAPH3 and β-actin, we generated complexes of DIAPH3 and β-actin using Alphafold2 multi-mer structure prediction[39,40] together with IDPConformerGenerator software in silico[41,42] (Fig. 3d and Supp Fig. 3a). This approach further confirmed that the DIAPH3 linker and the β-actin N-terminal residues are in close apposition while also suggesting they are flexible and capable of existing in multiple conformations (Fig. 3d and Supp Fig. 3b). These analyses position the linker region as the most likely region of the FH2 domain to interact with the variable actin N-termini. In addition, we compared the sequence conservation of the different FH2 sub-regions between DIAPH1 and DIAPH3. The region with the greatest sequence divergence was the linker region (Fig. 3c). Com-bined, these observations suggested that the linker region of the FH2 domain is a strong candidate to dictate actin isoform specificity in formins.

To test the model that the FH2 linker is the primary determinant of actin isoform specificity, we made further DIAPH1 and DIAPH3 hybrid proteins by swapping the FH2 linker regions between the two proteins. First, we used the mitochondrial targeting assays to assess the actin isoform specificity of the hybrid formins. While DIAPH1 generated both β- and γ-actin on mitochondria (Fig. 1e), DIAPH1 with the amino acids 687 to 705 of the DIAPH3 linker region replacing amino acids 820 to 842 of DIAPH1 (DIAPH1-3L) predominantly gener-ated β-actin on mitochondria (β-actin colocalization: MOC = 0.76 ± 0.17 compared to 0.06 ± 0.08 for empty vector control, $p = 1.45 \times 10^{-11}$; γ-actin colocalization: MOC = 0.28 ± 0.15 compared to 0.11 ± 0.12 for empty vector control, $p = 1.34 \times 10^{-4}$; Fig. 3e, g). In contrast, DIAPH3 with the amino acid 820 to 842 of the DIAPH1 linker region replacing amino acids 687 to 705 of DIAPH3 (DIAPH3-1L) generated both β- and γ-actin (β-actin colocalization: MOC = 0.65 ± 0.18 compared to 0.06 ± 0.08 for empty vector control, $p = 1.45 \times 10^{-11}$; γ-actin colocali-zation: MOC = 0.69 ± 0.19 compared to 0.11 ± 0.12 for empty vec-tor

control, $p = 1.74 \times 10^{-10}$; Fig. 3f, g). In vitro, actin polymerization assays were consistent with the cellular mitochondrial targeting data: DIAPH1-3L generated only β-actin whereas DIAPH3-1L generated both β- and γ-actins (Fig. 3h and Supp Fig. 2), indicating that actin isoform specificity is determined by the FH2 linker region and that the linker region of DIAPH3 specifies the preference for β-actin.

## The linker specifies actin isoform usage during cytokinesis
To determine if actin isoform networks can effectively substitute for one another during cytokinesis, we made use of our linker swap mutants to displace defined actin isoform networks during cytokinesis and then examined the consequences. We first established that the linker region in the context of full-length formins confers actin isoform specificity during cytokinesis, where β-actin is enriched at the furrow and γ-actin at the polar cortex[8,13]. We expressed full-length GFP-DIAPH1 and 3 linker swapped chimeras in HeLa cells and assessed the cellular distribution of β- and γ-actin filaments during cytokinesis. Stable HeLa cell lines expressing GFP-DIAPH1 and three variants under the control of doxycycline were generated to allow the expression of GFP fusion proteins at near endogenous levels (Supp Fig. 3c, d). Both linker swap chimeras, GFP-DIAPH1-3L (DIAPH1 with the DIAPH3 linker) and GFP-DIAPH3-1L (DIAPH3 with the DIAPH1 linker), localized as expected and coincident with endogenous DIAPH1 and 3. GFP-DIAPH1-3L was observed at the cell cortex similarly to DIAPH1[14], while GFP-DIAPH3-1L was enriched at the cytokinetic furrow as is DIAPH3[13] (Fig. 4a). In cells expressing GFP-DIAPH1-3L, which promotes β-actin polymerization, β-actin localized throughout the cell cortex and was no longer restricted to the furrow, while γ-actin remained around the cortex due to the activity of endogenous DIAPH1 (Fig. 4b, d).

Depletion of endogenous DIAPH1 by siRNA in cells expressing siRNA-resistant DIAPH1-3L led to the replacement of cortical γ-actin by β-actin, resulting in cells that had β-actin at both the cortex and furrow owing to the activities of the DIAPH1-3L chimera and endogenous DIAPH3 respectively (Fig. 4b, d). In contrast, in cytokinetic cells depleted of endogenous DIAPH3 and expressing a siRNA-resistant GFP-DIAPH3-1L chimera, γ-actin replaced β-actin at the furrow (Fig. 4c, d). It is noteworthy that the DIAPH3-1L chimera only produced γ-actin filaments at the plasma membrane in cellula compared to experiments utilizing just the C-terminal region of formins contain the DIAPH1 linker that generated both β- and γ-actin filaments. These observations suggest that in the context of the full-length protein, the DIAPH1 linker region is sufficient to specify a preference for the γ-actin isoform. Taken together, our observations suggest that during cyto-kinesis, both the β- and γ-actin networks can be relocalized using formins with altered specificity to modulate the actin isoform com-position at the cell cortex and cytokinetic furrow.

## Mis-localized actin isoforms disrupt cytokinesis
We first sought to determine if the mis-targeting of actin isoform networks affected successful cytokinesis as measured by the number of multi-nucleated cells, an indicator of cytokinetic failure. To

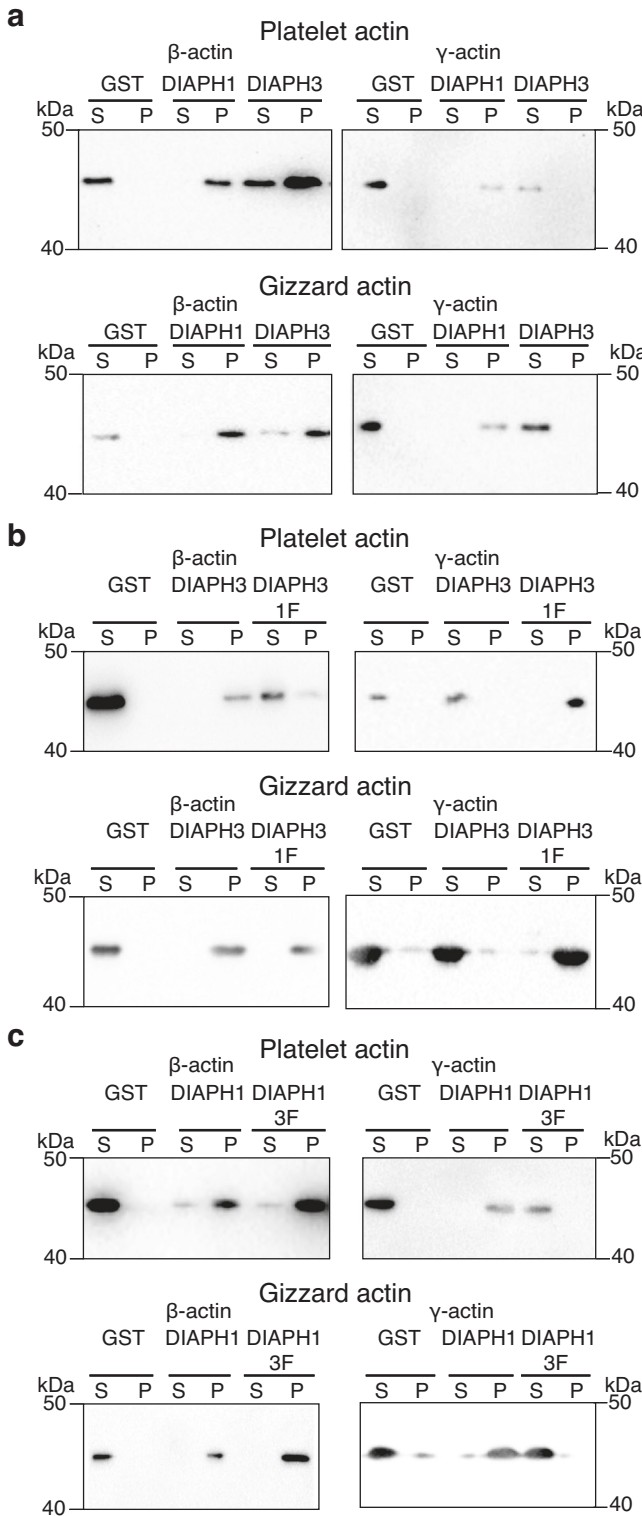

**Fig. 2 | The FH2 domain of DIAPH3 possesses β-actin isoform specificity in vitro.** **a** The CT of DIAPH1 and 3 fused to GST were incubated with actin purified from platelets (β-actin rich) or gizzard (γ-actin rich) and polymerized actin (P) separated from unpolymerized (S) by centrifugation. The fractions were then analyzed by immunoblotting using β- and γ-actin specific antibodies. Western blots presented are representative of three independent experiments. **b** As panel a except GST-DIAPH3 and chimeric GST-DIAPH3 containing the DIAPH1 FH2 domain (DIAPH3-1F) were used. Western blots presented are representative of three independent experiments. **c** As panel a except GST-DIAPH1 and chimeric GST-DIAPH1 containing the DIAPH3 FH2 domain (DIAPH1-3F) were used. Western blots presented are representative of three independent experiments.

determine if γ-actin could substitute for β-actin at the furrow, we assessed cytokinetic success in cells expressing GFP-DIAPH3-1L in the presence and absence of endogenous DIAPH3. Expression of GFP-DIAPH3-1L in the presence of endogenous DIAPH3 increased the proportion of multi-nucleated cells tenfold (from $4.63 \pm 1.34\%$ to $45.7 \pm 3.61\%$, $p = 8.97 \times 10^{-4}$; Fig. 5a), indicating that the presence of γ-actin in the furrow even in the presence of β-actin is detrimental to successful cytokinesis. Depletion of DIAPH3, which reduces β-actin at the furrow, caused severe cytokinetic failure ($57.8 \pm 2.47\%$ multinucleate cells in GFP-DIAPH3 cells depleted of DIAPH3 compared to $4.82 \pm 0.84$ for control siRNA-treated cells, $p = 2.22 \times 10^{-4}$; $68.2 \pm 3.63\%$ multinucleate cells in the GFP-DIAPH3-1L cell line compared to $4.63 \pm 1.34\%$, $p = 3.07 \times 10^{-4}$; Fig. 5a). While expression of GFP-DIAPH3 efficiently rescued the cytokinetic defects induced by DIAPH3 depletion ($7.29 \pm 1.50\%$ multi-nucleated cells, $p = 0.09$ in comparison with uninduced control siRNA-treated cells), the expression of GFP-DIAPH3-1L where γ-actin replaces β-actin in the furrow, failed to alleviate the cytokinetic phenotype ($50.7 \pm 4.74\%$ multi-nucleated cells, $p = 1.96 \times 10^{-3}$ in comparison with uninduced control siRNA-treated cells). These data suggest that γ-actin cannot functionally replace β-actin at the furrow to ensure successful cytokinesis.

To determine whether β-actin can functionally replace γ-actin at the cortex, we analyzed cytokinesis in cells expressing GFP-DIAPH1-3L that generate β-actin at the cortex, in the presence and absence of endogenous DIAPH1. Expression of GFP-DIAPH1-3L in the presence of endogenous DIAPH1 induced significant multinucleation ($33.2 \pm 2.97\%$ multinucleate cells compared to $4.21 \pm 1.05\%$ for control cells, $p = 1.41 \times 10^{-3}$), suggesting that the presence of β-actin at the polar cortex disrupts successful cytokinesis (Fig. 5b). Depletion of DIAPH1 disrupts cytokinesis ($32.4 \pm 2.32\%$ multi-nucleated cells in the GFP-DIAPH1 cell line) and was rescued by the expression of GFP-DIAPH1 ($4.68 \pm 1.14\%$ multi-nucleated cells, compared to $4.58 \pm 1.03\%$ for uninduced control cells; $p = 0.92$). In contrast, expression of GFP-DIAPH1-3L in DIAPH1-depleted cells further exacerbated the cytokinetic failure ($49.8 \pm 4.31\%$ of cells were multi-nucleated; $p = 1.86 \times 10^{-3}$ compared to $4.21 \pm 1.05\%$ for uninduced control cells; Fig. 5b). These data suggest that β-actin targeted to the cell cortex disrupts cytokinesis in the presence γ-actin and cannot substitute for the loss γ-actin to ensure successful cell division.

## Successful furrowing requires β-actin

We next sought to determine the events that become disrupted and lead to cytokinetic failure in cells with mis-localized actin isoforms. In cells with γ-actin rather than β-actin at the furrow, we measured furrow positioning using two metrics: the furrow ingression index and the furrow instability index[13] (Fig. 5c). The furrow ingression index assesses the symmetry of furrow ingression from each side of the cell equator: an index of one reflects symmetrical ingression and an index greater than one indicates asymmetric ingression. The furrow instability index indicates the position of the furrow: an index of one denotes a furrow position equidistant from the poles, while an index greater than one reflects cell furrowing at a position deviating from the midpoint of the cell. Depletion of DIAPH3 and the corresponding loss of β-actin at the furrow increased the ingression index from $1.13 \pm 0.0753$ in control cells to $1.90 \pm 0.18$ ($p = 1.45 \times 10^{-11}$, Fig. 5d) and increased the furrow instability index from $1.05 \pm 0.03$ to $1.94 \pm 0.18$ ($p = 1.45 \times 10^{-11}$, Fig. 5e). Furrow defects were rescued by the expression of GFP-DIAPH3 (ingression index = $1.10 \pm 0.057$, furrow instability index = $1.07 \pm 0.04$; $p = 0.29$ in both cases compared to respective controls; Fig. 5d, e), but not GFP-DIAPH3-1L that generates γ-actin at the furrow (ingression index = $1.62 \pm 0.07$, furrow instability index = $1.49 \pm 0.1$; $p = 1.45 \times 10^{-11}$ in both cases compared to respective controls; Fig. 5d, e). These data suggest that γ-actin cannot substitute for β-actin at the furrow to maintain correct furrow positioning and stability.

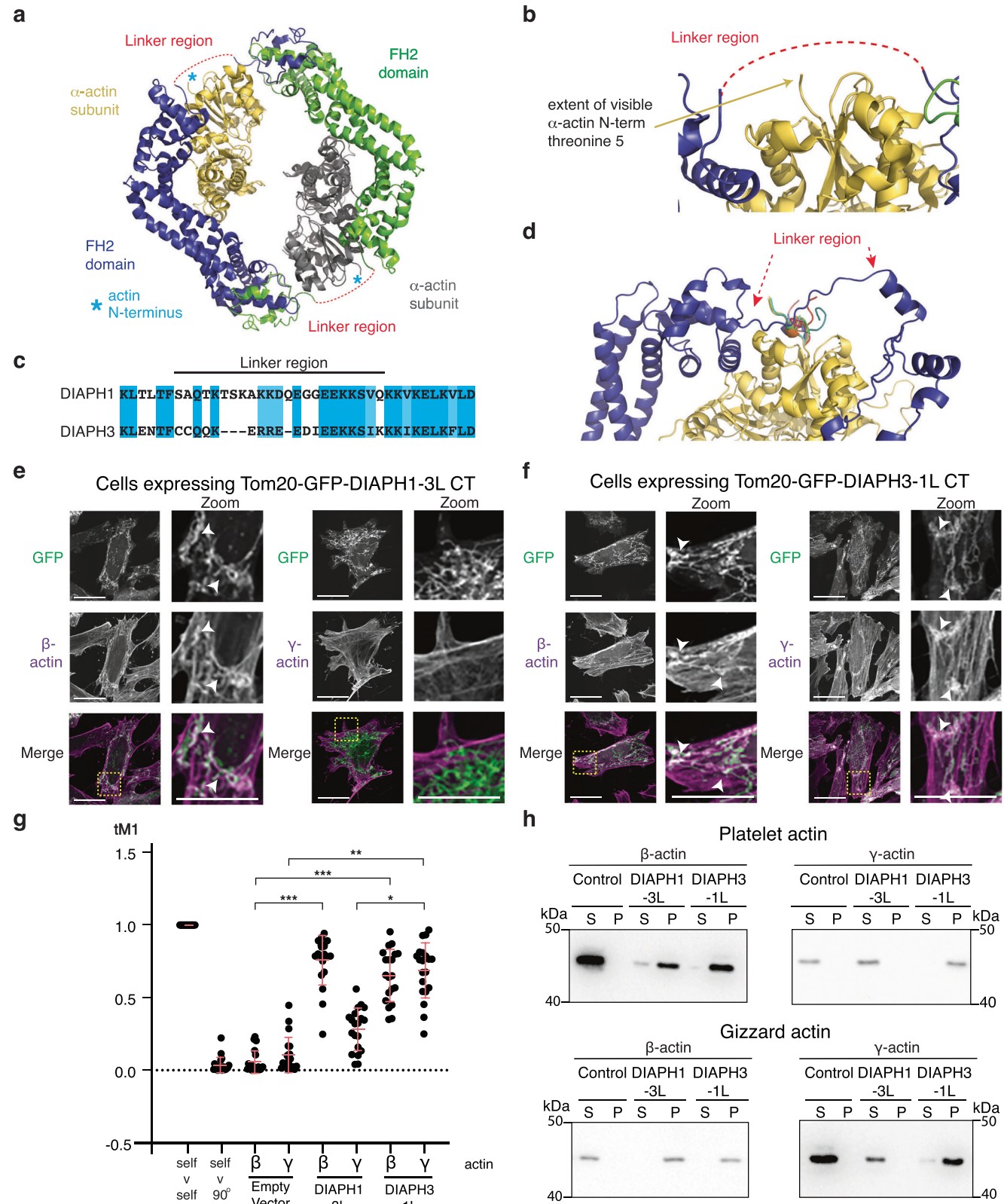

**c**

Linker region

DIAPH1 KLTLTFSAQTKTSKAKKDQEGGEEKKSVQKKVKELKVLD

DIAPH3 KLENTFCCQQK---ERRE-EDIEEKKSIKKKIKELKFLD

**e** Cells expressing Tom20-GFP-DIAPH1-3L CT

**f** Cells expressing Tom20-GFP-DIAPH3-1L CT

**g**

**h**

We have previously shown that cortical γ-actin has a role in regulating furrow ingression[14]. As expected, the depletion of DIAPH1 and the corresponding loss of cortical γ-actin disrupted furrow organization (ingression index of $1.11 \pm 0.06$ in control cells vs $2.05 \pm 0.41$, $p = 1.45 \times 10^{-11}$ in DIAPH1 depleted cells (Fig. 5f) and an instability index of $1.06 \pm 0.03$ in control cells vs $1.77 \pm 0.19$, $p = 1.45 \times 10^{-11}$ in DIAPH1 depleted cell, Fig. 5g). The ingression and instability indices were reduced to near control levels by expression of GFP-DIAPH1 in DIAPH1 siRNA-treated cells (Fig. 5f, g). In contrast, expression of GFP-DIAPH1-3L that generated β- rather than γ-actin at the cortex only partially rescued the furrow ingression index ($1.46 \pm 0.30$, $p = 1.34 \times 10^{-7}$, Fig. 5f) and did not rescue the furrow instability index ($1.67 \pm 0.15$; $p = 1.45 \times 10^{-11}$, Fig. 5g). These data suggest that β-actin cannot substitute for γ-actin at the cortex to maintain correct furrow positioning and stability.

**Fig. 3 | The linker region of the DIAPH3 FH2 domain determines DIAPH3 β−actin isoform specificity. a** View of the co-crystal structure of a dimer of FMNL3 FH2 domains (dark blue and green) with two actin subunits (yellow and gray). Positions of the missing linker region and actin N-terminus are inferred in red and with a blue star, respectively. **b** Close-up view of the area the two missing regions are inferred to occupy. **c** Sequence alignment of the linker regions of DIAPH1 and 3. **d** Top ranked Alphafold2 model of the complex between DIAPH3 FH2 domain (blue) and β-actin (yellow). An ensemble of 10 Local Disordered Region Sampling (LDRS)-generated conformers of the β-actin N-terminus are shown in a color spectrum, a single conformer of the FH2 linker is shown. **e** Micrographs of HeLa cells expressing a Tom20-GFP-DIAPH1-CT fusion protein that contains the linker sub-domain of DIAPH3 (green), fixed and stained for β- or γ-actin (magenta). **f** Micrographs of HeLa cells expressing a Tom20-GFP-DIAPH3-CT fusion protein containing the linker sub-domain of DIAPH1 (green), fixed and stained for β- or γ-actin (magenta). For panels e and f: scale bars represent 10 μm; yellow boxed regions are enlarged in adjoining zoom panels; white arrowheads indicate instances of actin-mitochondria colocalization; micrographs are representative of three independent experiments. **g** Relative colocalization determined by Manders' overlap coefficient (tM1) of β- and γ-actin to mitochondria in HeLa cells expressing Tom20-GFP fused to the CT domains of DIAPH1 and DIAPH3 containing the linker sub-domain of either DIAPH1 (1 L) or DIAPH3 (3 L). Data were presented as mean (solid bar) ± SD (error bars). $n = 20$ cells analyzed per condition across three independent experiments. $*p = 4.18 \times 10^{-8}$, $**p = 1.74 \times 10^{-10}$, $***p = 1.45 \times 10^{-11}$ as determined by two-sided Mann−Whitney non-parametric tests. **h** The CT of DIAPH1 and 3 containing the linker sub-domain of either DIAPH1 (1 L) or DIAPH3 (3 L) fused to GST were incubated with actin purified from platelets (β-actin rich) or gizzard (γ-actin rich) and polymerized actin (P) separated from unpolymerized (S) by centrifugation. Fractions were analyzed by immunoblotting using actin isoform-specific antibodies. The blots presented are representative of three independent experiments.

To gain further insight into the requirement for distinctly localized actin isoform networks, we monitored the dynamics of furrow ingression in cells with either wild-type or mis-localized actin isoform networks using timelapse phase contrast microscopy. Depletion of DIAPH1 (and thus depleting cortical γ-actin) reduced furrow ingression rates from $0.78 \pm 0.09$ μm/min to $0.54 \pm 0.17$ μm/min ($p = 0.03$, Fig. 5h). The reduced furrow ingression rate in DIAPH1 depleted cells was reversed by expression of WT GFP-DIAPH1 ($1.04 \pm 0.07$ μm/min, $p = 7.94 \times 10^{-3}$). Expression of GFP-DIAPH1-3L that generates β-actin at the cortex increased the furrow ingression rate in DIAPH1-depleted cells to a rate comparable to the expression of GFP-DIAPH1 ($1.05 \pm 0.13$ μm/min, $p = 7.94 \times 10^{-3}$; Fig. 5h). These observations suggest that the role of cortical actin in furrow ingression is independent of the actin isoform network generated at the cortex. In contrast, the actin isoform at the furrow is a critical parameter for cytokinetic success: depletion of β-actin at the furrow reduced the furrow ingression rate from $0.88 \pm 0.06$ μm/min to $0.78 \pm 0.05$ μm/min ($p = 0.0317$), and the ingression rate was not rescued by the presence of γ-actin (targeted to the furrow through the expression of DIAPH3-1L, ingression rate of $0.73 \pm 0.08$, $p = 0.38$ compared to DIAPH3 depleted cells; Fig. 5i).

Mis-localized actin isoform networks induced additional phenotypes in dividing cells, notably an increase in plasma membrane blebbing and a reduction in cell elongation rates during anaphase B. Regarding blebbing, we observed distinct effects in the furrow and polar regions of the dividing cell (Fig. 6a). Blebbing at the site of the ingressing furrow was rarely observed across most conditions. However, blebs were observed in cells where β-actin at the furrow was replaced by γ-actin in cells expressing GFP-DIAPH3-1L in the absence of endogenous DIAPH3 ($5.60 \pm 1.36$ furrow blebs compared to $0.80 \pm 0.78$ for DIAPH3 depleted cells, $p = 1.08 \times 10^{-5}$; Fig. 6a–c). In contrast, blebs at the poles of dividing cells were routinely observed as expected[43]. Depleting either endogenous DIAPH1 to disrupt cortical γ-actin or endogenous DIAPH3 to disrupt β-actin at the furrow increased the number of polar blebs (from $4.13 \pm 1.05$ to $14.2 \pm 3.16$ for DIAPH1 depletion, $p = 4.57 \times 10^{-5}$; and from $5.45 \pm 1.50$ to $11.1 \pm 1.92$ for DIAPH3 depletion, $p = 8.51 \times 10^{-6}$; Fig. 6b, c). This increase in polar blebbing was reversed by the expression of GFP-DIAPH1 or GFP-DIAPH3 ($6.80 \pm 1.72$ polar blebs for GFP-DIAPH1, $p = 3.25 \times 10^{-5}$ compared to DIAPH1 depleted cells; and $6.70 \pm 1.85$ polar blebs for GFP-DIAPH3, $p = 2.27 \times 10^{-4}$ compared to DIAPH3 depleted cells; Fig. 6b, c), but not by the expression of DIAPH1-3L or DIAPH3-1L that mis-localized actin isoform networks ($13.1 \pm 1.97$ polar blebs for DIAPH1-3L, $p = 0.13$ compared to DIAPH1 depleted cells; $11.5 \pm 2.06$ polar blebs for DIAPH3-1L, p = 0.09 compared to DIAPH3 depleted cells; Fig. 6b, c).

Finally, we observed that the actin cytoskeleton is essential for cell elongation during anaphase B and that γ-actin is the dominant actin network involved in anaphase B cell elongation rate, as depleting DIAPH1 reduced the rate of cell elongation from $0.44 \pm 0.04$ to $0.3 \pm 0.05$ μm/min ($p = 4.58 \times 10^{-4}$; Fig. 6d). Cell elongation defects were rescued upon the expression of GFP-DIAPH1 (elongation rate of $0.47 \pm 0.09$ μm/min, $p = 0.99$ compared to control cells), but not by the GFP-DIAPH1-3L chimera that generates β-actin at the cortex instead of γ-actin (elongation rate = $0.35 \pm 0.05$ μm/min compared to $0.33 \pm 0.05$ μm/min for GFP-DIAPH1-3L cells depleted of DIAPH1, $p = 0.84$). Conversely, the loss of β-actin at the furrow due to DIAPH3 depletion led to only minor reductions in cell elongation rate (Fig. 6d). These data suggest that cell elongation requires a cortical γ-actin network and further support the model that β-actin and γ-actin have distinct properties independently required for different cellular processes.

### β-actin dictates nm myoII localization

To further understand the molecular pathways reliant on the different actin isoform networks, we examined the localization of different cytoskeletal factors known to act immediately upstream or downstream of actin during furrow formation and ingression. Anillin, a cytokinetic scaffolding factor involved in activating DIAPH3[13], was unaffected by swapping actin isoforms either at the furrow or at the cortex (Supp Fig. 4A). Likewise, no changes were observed to the microtubule cytoskeleton (Sup Fig. 4b). In contrast, the localization of phosphorylated myosin light chain, a reporter for activated non-muscle myosin II (nm myoII), was found to be dependent on β-actin localization. In control cells, the phospho-myosin light chain is found at the β-actin-enriched ingressing furrow (Fig. 7a–c). Replacing β-actin with γ-actin at the furrow (by expressing GFP-DIAPH3-1L in cells depleted of endogenous DIAPH3), led to the depletion of phospho-myosin light chain from the furrow membrane, instead concentrated at the spindle midzone (Fig. 7a–c). Interestingly, moving β-actin from the furrow to the polar cortex in cells expressing GFP-DIAPH1-3L redistributed the phosphorylated myosin light chain from the furrow to the entire plasma membrane around the cortex (plasma membrane phospho-myosin light chain fluorescence intensity furrow:pole ratio = $0.53 \pm 0.09$ compared to $1.70 \pm 0.47$ in GFP-DIAPH1 expressing cells, $p = 7.94 \times 10^{-3}$; Fig. 7a, b), indicating that activated nm myoII localization during cytokinesis is dependent upon the positioning of the β-actin network

### β-actin maintains RhoA at the furrow

The localization of RhoA-GTP, the active form of RhoA, to the future site of furrow ingression, is considered a crucial early event driving cytokinesis[44,45]. However, while RhoA-GTP initiates a key signal transduction cascade needed to initiate the early events of cytokinesis, it appears to be influenced by feedback pathways involving both upstream factors, such as its guanine nucleotide exchange factor Ect2[30] and downstream factors, including actin[46]. Therefore, we assessed the effect of mis-localizing actin isoform networks on RhoA and its regulators during the early stages of cytokinesis. In control cells

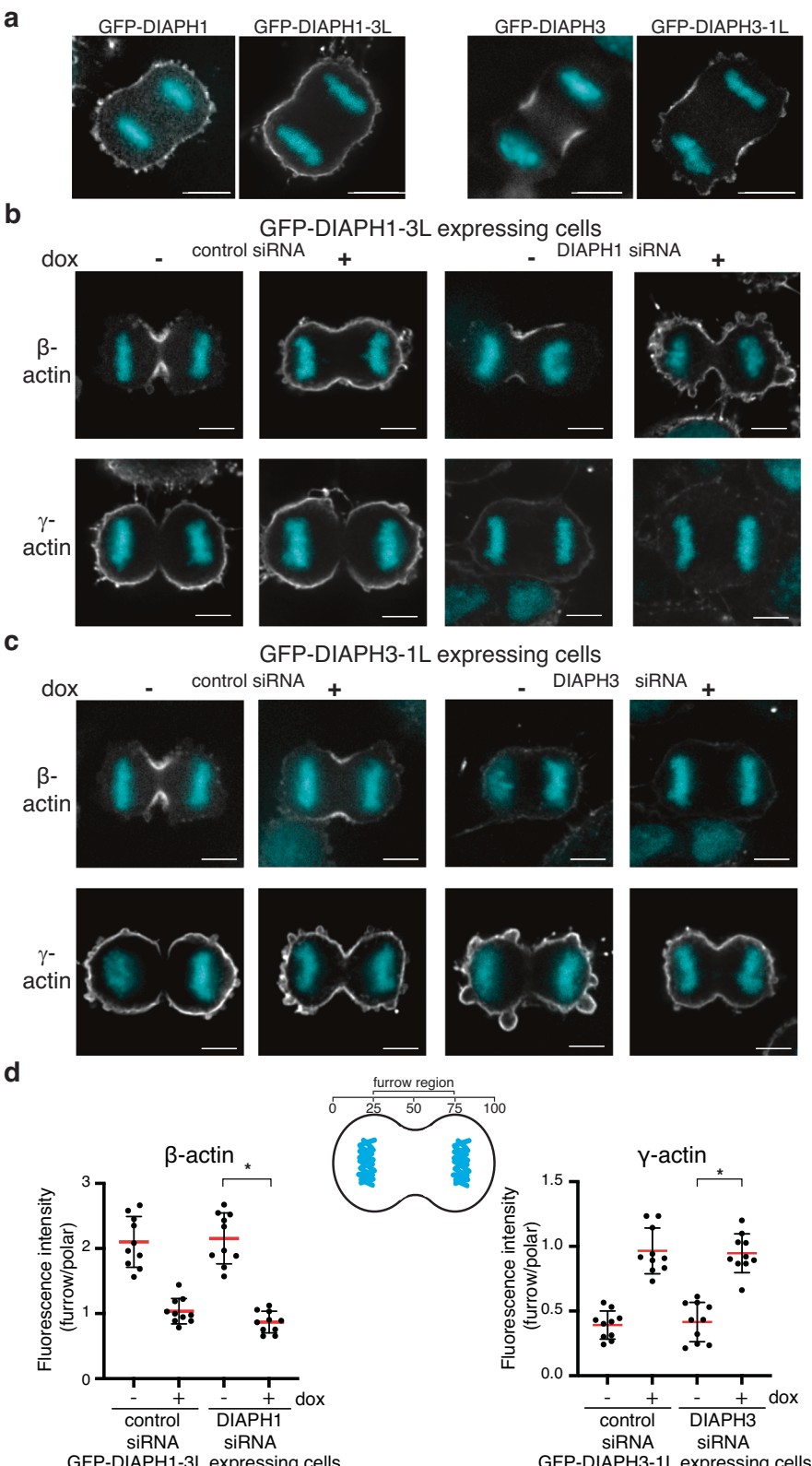

**Fig. 4 | The linker region of the FH2 domain determines actin isoform specificity of DIAPH1 and 3 in cellula. a** Micrographs of HeLa cells expressing GFP-DIAPH1, GFP-DIAPH1 containing the DIAPH3 linker (GFP-DIAPH1-3L), GFP-DIAPH3, and GFP-DIAPH3 containing the DIAPH1 linker (GFP-DIAPH3-1L). The presented micrographs are representative of three independent experiments. **b** Micrographs of HeLa cells expressing GFP-DIAPH1-3L in the presence or absence of endogenous DIAPH1, stained with β- or γ-actin isoform-specific antibodies. The presented micrographs are representative of three independent experiments. **c** Micrographs of HeLa cells expressing GFP-DIAPH3-1L in the presence or absence of endogenous DIAPH3, stained with β- or γ-actin isoform-specific antibodies. The presented micrographs are representative of three independent experiments. **d** Top: Cartoon illustrating the subdivision of dividing cells into furrow and polar regions. Bottom: Quantitation of the intensity of the indicated actin isoform staining within the furrow and polar regions of anaphase cells under different conditions. $n = 10$ cells analyzed across three independent experiments. Data were presented as mean (red bars) ± SD. *$p = 1.083 \times 10^{-5}$ as determined by two-sided Mann–Whitney non-parametric tests.

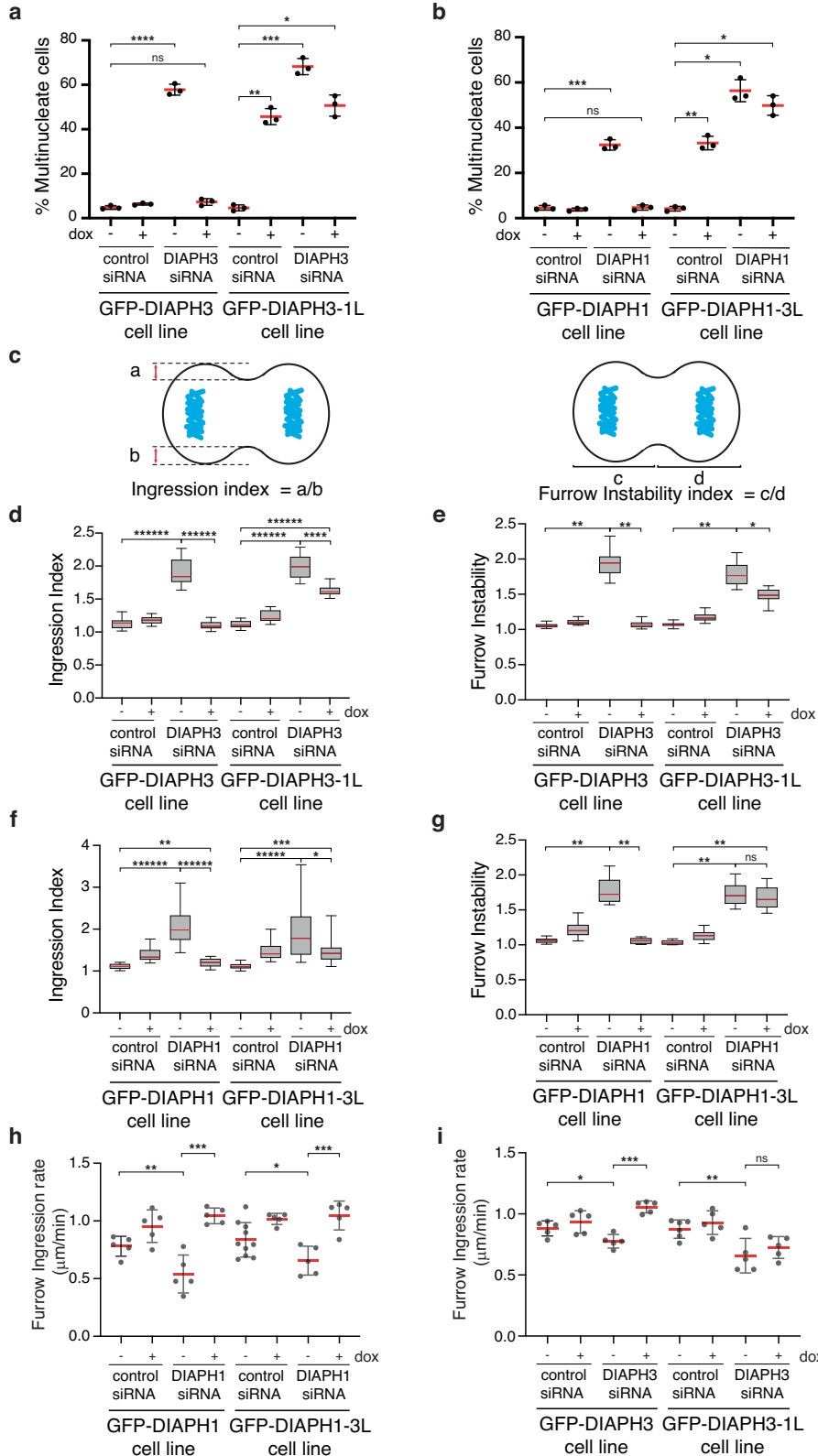

where the β-actin network was at the furrow and the γ-actin network at the cortex, RhoA and Ect2 were enriched at the furrow as expected (Fig. 7d, e). However, in cells where the β-actin was replaced by a γ-actin network at the furrow (cells depleted of DIAPH3, but expressing DIAPH3-1F or DIAPH3-1L), neither RhoA nor Ect2 localized to the furrow (Fig. 7d, e). Interestingly, the localization of components of the centralspindlin complex, RacGAP1 and MKLP1, which target Ect2 to the

furrow, remained unaffected in these cells (Supp Fig. 4c, d). However, upon the relocalization of β-actin throughout the cortex (including the furrow) in cells depleted of DIAPH1 and expressing both GFP-DIAPH1-3L and endogenous DIAPH3, RhoA still localized to the furrow. Collectively, these data indicate that the presence of β-actin at the furrow is essential to maintain the presence of both activated nm myoII and active RhoA there, thereby ensuring successful cytokinesis.

**Fig. 5 | Correct positioning of β- and γ-actin networks is required for successful cytokinesis. a** Quantitation of multinucleate cells formed upon expressing GFP-DIAPH3 or GFP-DIAPH3-1L with or without endogenous DIAPH3. For panels **a**, **b**: data were presented as mean (solid bar) ± SD (error bars); $n = 900$ cells scored per condition across three independent experiments. "ns" = 0.085, $*p = 1.96 \times 10^{-3}$, $**p = 8.97 \times 10^{-4}$, $***p = 3.07 \times 10^{-4}$, $****p = 2.22 \times 10^{-4}$ as determined by two-sided Mann–Whitney non-parametric tests. **b** Quantitation of multinucleate cells formed upon expressing GFP-DIAPH1 or GFP-DIAPH1-3L with or without endogenous DIAPH1. "ns" = 0.92, $*p = 1.86 \times 10^{-3}$, $**p = 1.41 \times 10^{-3}$, $***p = 5.21 \times 10^{-4}$ as determined by two-sided Mann–Whitney non-parametric tests. **c** Cartoons illustrating derivation of ingression and furrow instability indices. **d** Ingression indices of different cell lines with or without DIAPH3 and DIAPH3-1L expression. For panels **d**–**g**: $n = 20$ cells analyzed across three independent experiments; red bars denote the mean, the lower bound of boxes represent the 25th percentile, the upper bound of boxes represent the 75th percentile, and whiskers denote minima and maxima. For panels

**d, f**: $*p = 4.27 \times 10^{-3}$, $**p = 3.21 \times 10^{-3}$, $***p = 1.34 \times 10^{-7}$, $****p = 1.02 \times 10^{-10}$, $*****p = 5.80 \times 10^{-11}$, $******p = 1.45 \times 10^{-11}$ determined by two-sided Mann–Whitney non-parametric tests. **e** Furrow stability indices of different cell lines with or without DIAPH3 and DIAPH3-1L expression. For panels **e**, **g**: "ns" $p = 0.34$, $*p = 1.02 \times 10^{-10}$, $**p = 1.45 \times 10^{-11}$ determined by two-sided Mann–Whitney non-parametric tests. **f, g** Ingression indices of cells treated as indicated. **h, i** Furrow ingression rates of cells treated as indicated. Data were presented as mean (red bars) ± SD. $n = 5$ cells analyzed for all conditions across three independent experiments, except "GFP-DIAPH1-3L control siRNA -DOX" and "GFP-DIAPH3-1L control siRNA -DOX" where $n = 10$ and 6 respectively across 5 and 4 independent experiments. For panel **h**: $*p = 0.04$, $**p = 0.032$, $***p = 7.94 \times 10^{-3}$ determined by two-sided Mann–Whitney non-parametric tests. For panel **i**: "ns" = 0.38, $*p = 0.032$, $**p = 0.030$, $***p = 7.94 \times 10^{-3}$ determined by two-sided Mann–Whitney non-parametric tests.

## Discussion

We report that during cell division, the remodeling of the plasma membrane depends on specialized β- and γ-actin isoform networks that localize to the cytokinetic furrow and cell cortex, respectively. Importantly, we show that these actin isoforms uniquely contribute to cytokinesis and cannot substitute for one another. Our observations reveal that distinct actin isoform networks dictate the localization of the key cytokinetic factors, RhoA, Ect2, and nm myoII. Consequently, the actin isoform networks make a significant contribution to establishing the different biochemical environments at the plasma membrane needed to drive the spatially restricted membrane remodeling events that underlie successful cell division.

The variation in actin isoforms primarily derives from the divergent amino acid sequences of the extreme N-termini. This diversity appears to be an evolutionary event associated with the emergence of the plant and animal kingdoms. For example, the yeasts *S. cerevisiae* and *S. pombe* have a single actin isoform, whereas humans have seven. Yeast also have only a limited number of formins, each with defined functions and subject to different regulatory pathways[5,47,48]). In contrast, the explosion in the number of actin isoforms in plants and animals corresponds with that of formins (14 in humans). As there is increasing evidence of actin isoform-specific functions (this study and others reviewed in ref. [6]), this diversification of formins and other actin nucleators suggests that actin isoforms and nucleators co-evolved to generate actin isoform networks with specific functions that are subject to divergent regulatory pathways.

It is intriguing that a change of only four amino acids at the extreme N-terminus of actin dictates diverse cellular roles. While the substitution of three aspartic for glutamic acids, and isoleucine for its isomer valine are the most conservative that can be made, a recent cryoEM study revealed differences in the orientation of the N-termini of actin isoforms[49] suggesting that different actin N-termini have the capacity to interact differentially with downstream partners.

The formin family of proteins are defined by their conserved FH2 domain. In this work, we demonstrate that the FH2 domains of DIAPH1 and DIAPH3 confer the unique actin isoform preference of each formin, raising the possibility that the FH2 domains of other formins may do likewise. Whilst no direct experimental or structural information exists, in silico modeling predicts that salt bridges can form between the N-terminus of α-actin and the linker region of mouse formin DIAPH1, a feature that holds true for the *S. cerevisiae* formin Bni1 and the *S. pombe* formin Cdc12[38]. In a comparative analysis of DIAPH1 and DIAPH3 FH2 linker sequences, no conserved amino acid sequence motifs that account readily for actin isoform specificity could be identified (Supp Fig. 5). The DIAPH1 linker is 2–4 amino acids longer compared to DIAPH3, while the DIAPH1 linker carries a slightly basic net charge compared to the slightly acidic character of the DIAPH3 linker sequence. In addition, our modeling studies suggest that the FH2 linker region itself is flexible and able to exist in different conformations that perhaps allow the linker to tolerate different actin

N-termini conformations found between the various actin isoforms. It is tempting to speculate that constraining some linkers to a smaller set of conformations, perhaps due to their length or the charge distribution along it, could serve to restrict the actin isoforms it interacts with. When combined with the diversity of the actin N-termini and the varied molecular space that they can occupy, this may bring optimal and suboptimal residues into such an arrangement as to establish actin isoform preferences for a given formin. Further factors are likely to contribute to formin specificity, as in vitro, the constitutively active DIAPH1 C-terminal fragment generated both β- and γ-actin homopolymeric filaments, while in cellula full-length DIAPH1 at the cell cortex produces only γ-actin filaments during cytokinesis. Thus, while the sequence of the FH2-linker accounts for a great deal of a formin's actin isoform preference, it is unlikely to give a complete account of it. Whether these differences manifest themselves through the rate of filament nucleation and polymerization or whether specific post-translation modifications or additional cellular factors play a role remains to be determined.

Our re-positioning of the β- and γ-actin networks severely disrupted cytokinesis demonstrating that the β- and γ-actin networks play non-redundant roles in carrying out successful cell division. Crucially, replacing β-actin at the furrow with γ-actin destabilized the furrow and reduced the rate of furrow ingression, while a γ-actin network at the cortex was required for cell elongation during anaphase. Interestingly, an actin isoform-independent need for cortical actin in furrow ingression was observed. Indeed, disrupting the cortical actin had a far greater effect on furrow ingression than disrupting actin at the furrow itself. These observations further support the model that the coordinated modulation of activities at the poles plays an important role in ensuring furrow ingression and successful cytokinesis[14,50–54], and that events at the furrow have a role in stabilizing furrow positioning[13,43]. Our data suggest that at the furrow, actin interactors that distinguish between the β- and γ-actin N-termini are required, whereas the cortical actin interactors contact parts of the actin filament common to both β- and γ-actin.

The divergent N-terminal residues of the actin isoforms extend outwards from the filament making this region accessible to potential cytoplasmic interactors[49,55]. We observed that the localization of the phosphorylated myosin light chain, which marks activated nm myoII, was downstream of β-actin during cytokinesis. In WT cells, β-actin concentrates at the furrow and nm myoII is likewise observed there. Substituting γ-actin for β-actin at the furrow disrupts nm myoII localization, suggesting nm myoII is recruited and/or stabilized at the furrow in a β-actin dependent manner. Consistent with this, when the β-actin network was ectopically extended throughout the cortex of a dividing cell, nm myoII localization mirrored that of β-actin, suggesting that the β-actin network at the furrow specifically positions nm myoII, and that its activity contributes to both stabilizing the furrow in the equatorial region of the cell and driving furrow ingression.

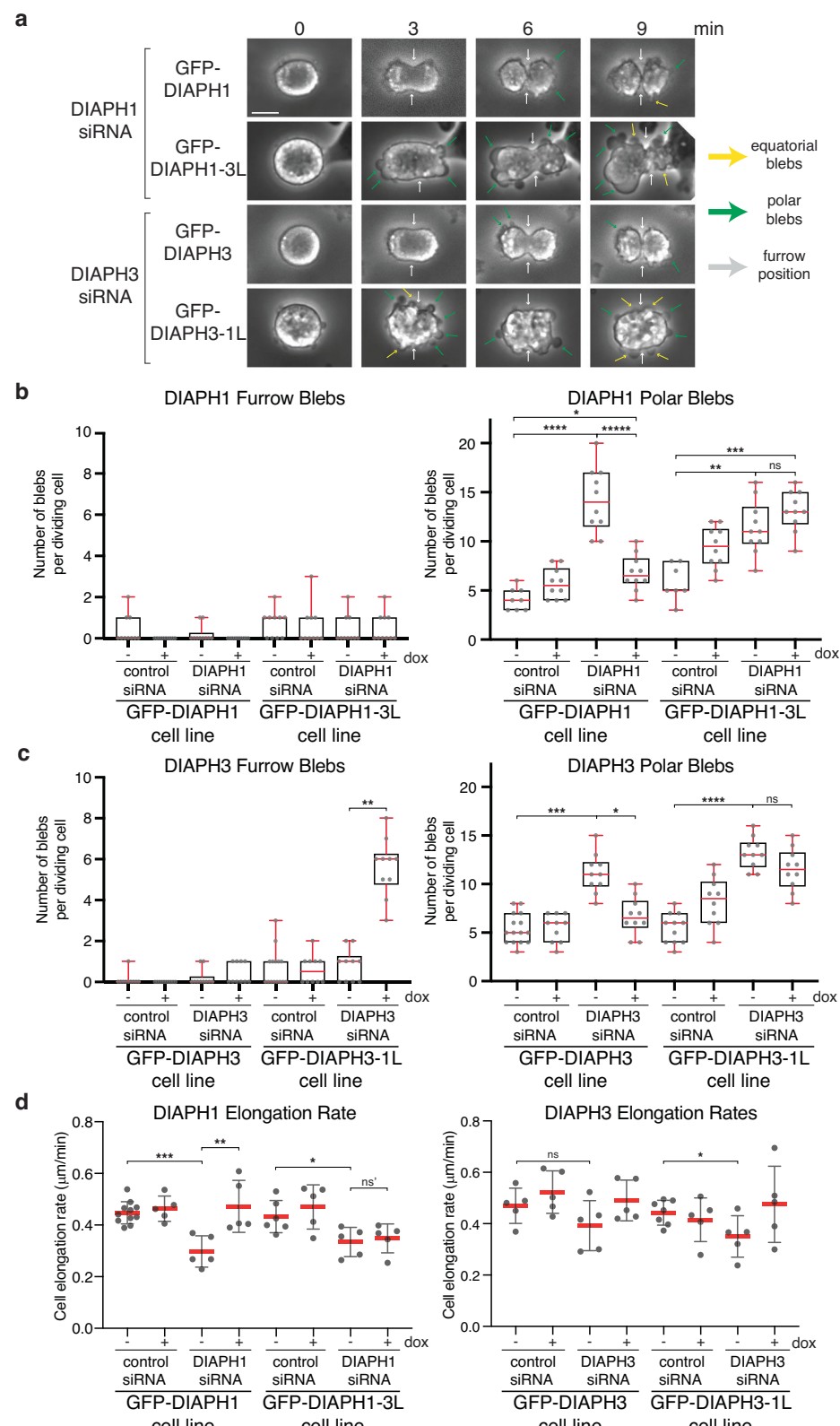

Our observations suggest that the actin N-terminus exerts considerable influence over nm myoII function. It has long been known that the N-terminus of actin isoforms interacts directly with myosin[56–59], work that has been further confirmed in structural studies[49,55]. While nm myoII isoforms have greater ATPase activity in vitro in the presence of β- and γ-actin compared to their activity on α-actins, little difference in activity was observed between β- and γ-actin except for the nm myoIIC isoform[60]. In contrast, myosin7A is potently activated by γ-actin compared to other actin isoforms, suggesting that at least some myosins exhibit actin isoform preferences in these assays. Therefore, it is possible that the unique conformations of the divergent actin isoform N-termini have distinct interactions with specific myosins to influence their activity and function.

**Fig. 6 | Mis-positioning of actin isoform networks during anaphase disrupts cell elongation and induces blebbing. a** Phase contrast images from timelapse movies of dividing cells expressing DIAPH1 and 3 wildtype and linker region mutants. $t = 0$ defined as first frame following anaphase onset. The scale bar represents 10 μm. **b** Quantitation of the number of blebs occurring at the furrow or poles of dividing cells expressing different DIAPH1 variants with or without endogenous DIAPH1. $n = 10$ cells scored per condition for both furrow and polar blebs across three independent experiments, except for "GFP-DIAPH1 control siRNA -DOX" and "GFP-DIAPH1-3L control siRNA -DOX" where $n = 8$ and 7, respectively. Red bars denote the median, lower bound of boxes represent the 25th percentile, upper bound of boxes represent the 75th percentile, and whiskers represent maxima and minima. "ns" = 0.13, *$p = 2.03 \times 10^{-3}$, **$p = 5.14 \times 10^{-4}$, ***$p = 1.03 \times 10^{-4}$, ****$p = 4.57 \times 10^{-5}$, *****$p = 3.25 \times 10^{-5}$ determined by two-sided Mann−Whitney non-parametric tests. **c** Quantitation of furrow and polar blebs in dividing cells expressing different DIAPH3 variants with or without endogenous DIAPH3. $n = 10$ cells scored for furrow

and polar blebs across three independent experiments, except for "GFP-DIAPH3 control siRNA -DOX" for polar blebs and 'GFP-DIAPH3-1L control siRNA -DOX' for furrow blebs where $n = 15$. Red bars denote the median, lower bound of boxes represent the 25th percentile, upper bound of boxes represent the 75th percentile, and whiskers represent minima and maxima. "ns" = 0.085, *$p = 2.27 \times 10^{-4}$, **$p = 1.08 \times 10^{-5}$, ***$p = 8.51 \times 10^{-6}$, ****$p = 3.06 \times 10^{-7}$ determined by two-sided Mann−Whitney non-parametric tests. **d** The pole-to-pole cell elongation rates of dividing cells expressing indicated constructs with or without endogenous DIAPH1 or DIAPH3. $n = 5$ cells analyzed per condition across three independent experiments, except for "GFP-DIAPH1 control siRNA -DOX", "GFP-DIAPH1-3L control siRNA -DOX", and "GFP-DIAPH3-1L control siRNA -DOX" where $n = 11$, 6, and 7, respectively across five independent experiments. Data were presented as mean (red bars) ± SD. ns = 0.84, ns = 0.22, *$p = 0.03$, **$p = 7.94 \times 10^{-3}$, ***$p = 4.58 \times 10^{-4}$ determined by two-sided Mann−Whitney non-parametric tests.

General models describing actin network dynamics place small G-proteins like RhoA upstream of actin polymerization, controlling the activation of actin nucleators to generate the actin network itself. However, recent observations, including those reported in this study, suggest that actin networks also feedback upstream of the signaling cascade to moderate G-protein function. On the excitable cortex of *Xenopus* oocytes and sea urchin embryonic cells, waves of actin traverse great distances[46,61]. Ahead of the actin is a wave of RhoA and actin nucleators. The leading edge of the actin wave appears to deactivate the wave of RhoA ahead of it, emphasizing a feedback mechanism from actin to RhoA[46]. Whilst these waves are far larger than the cytokinetic furrow of mammalian tissue culture cells, they may provide a mechanistic template to understand cytokinesis. As relocalizing β-actin around the cortex does not relocalize RhoA, β-actin is unlikely to be a direct effector of RhoA. Rather, as suggested from the studies on actin waves, β-actin is more likely to influence the activity of a RhoA regulator. Indeed, we observed that the equatorial localization of the RhoGEF Ect2 was disrupted by the relocalization of β-actin around the cortex. Such a mechanism may allow the fine tuning of the "Rho zone" during the changing environment of the ingressing furrow to attenuate the different Rho-dependent pathways to ensure furrowing is successful.

Our findings demonstrate the crucial roles of distinct actin isoform networks in successful cytokinesis, with each network having unique functions that cannot be replaced by others. These independent networks, controlled by nucleators like formins, exhibit preferences for specific actin isoforms. This diversity in actin isoform networks leads to the generation of platforms with distinct biochemical properties, enabling specific activities to be carried out at targeted regions within the cell. Further exploration of this diversity is likely to uncover its contribution to numerous cellular functions.

## Methods

### Cloning
pGEX-6P-2-DIAPH1-CT and pGEX-6P-2 DIAPH3-CT plasmids along with full-length GFP-DIAPH1 (using the human cDNA) and DIAPH3 (using the mouse cDNA) in the pcDNA5 vector (Invitrogen) were described previously[13,14,62]. Chimeric full-length DIAPH1 and DIAPH3 DIAPH1-CT and DIAPH3-CT were generated in like manner, first by PCR amplification of the FH2 and linker domains using oligonucleotides (IDT) described in Supplementary Table 1. Stitched chimeric formins were cloned into the plasmids pGEX-6P2, pcDNA5, or a modified pEGFP vector downstream and in frame with a TOM20Tm-GFP fusion protein (a gift from Dr. P. Kim, Hospital for Sick Children, Toronto) using the InFusion system (Takara).

### Protein expression and purification
Recombinant proteins were purified from BL21 *E. coli* cells transformed with plasmids containing GST fusion proteins, grown in LB media at

37 °C to an optical density of 0.6 at $A_{600}$. Recombinant protein expression was induced by the addition of 1 mM isopropyl β-ᴅ-1-thio-galactopyranoside (IPTG) and further incubated at 16 °C overnight. Cells were harvested by centrifugation and stored at −80 °C. Cell pellets were thawed and resuspended in 25 mM HEPES, pH 7.5, 250 mM NaCl, 100 mM KCl, 0.5 mM β- mercaptoethanol, 1 mM PMSF, and lysed by sonication. The lysates were cleared by centrifugation at 10,000×$g$ for 30 min at 4 °C, and the supernatant was applied to glutathione beads (Invitrogen). The glutathione beads were washed with 10 column volumes of column buffer (CB) containing 25 mM HEPES, pH 7.5, 250 mM NaCl, 100 mM KCl, 0.5 mM β- mercaptoethanol, 1 mM PMSF, 0.1% (v/v) Triton X-100, prior to elution of GST fusions in CB containing 10 mM glutathione.

### Immunofluorescence and microscopy analysis
To visualize β- and γ-actin in HeLa cells expressing GFP-DIAPH1, GFP-DIAPH3, GFP-DIAPH1-3F, and GFP-DIAPH3-1F, cells were fixed with 3.7% formaldehyde at 37 °C for 30 min followed by a methanol-post fixation treatment for 15 min at −20 °C. To visualize DIAPH1, DIAPH3, activated nm myoII, RacGAP1, as well as WT and chimeric GFP-DIAPH constructs, cells were fixed with 3.7% formaldehyde at room temperature for 10 min and permeabilized with 0.2% Triton X-100 for 10 min. To visualize RhoA and anillin, cells were fixed with 10% TCA in a Cytoskeleton Fixation Buffer (CFB) containing 10 mM MES pH 6.1, 320 mM sucrose, 138 mM KCl, 3 mM MgCl₂, 2 mM EGTA at 4 °C for 20 min. Cells were then permeabilized with 0.2% Triton X-100 in GPBS (PBS containing 50 mM glycine) for 2 min at 4 °C then quenched with GPBS for 20 min at room temperature. To visualize Ect2, MKLP1, and α-tubulin, cells were fixed with ice-cold methanol for 10 min at −20 °C. Following fixation, coverslips were washed with PBS prior to blocking with 3% BSA in PBS at room temperature for 1 h. Cells were stained with indicated antibodies (see Supplementary Table 3) for either 1 h at room temperature or overnight at 4 °C. Coverslips were washed with PBS prior to incubation with secondary antibodies (see Supplementary Table 3) for 1 h at room temperature. Coverslips were then stained with Hoechst (Sigma-Aldrich) for 10 min to visualize DNA, before mounting on glass slides with Mowiol (Polyvinyl alcohol 4-88, Fluka). Samples were visualized using a Perkin Elmer Ultraview spinning disk confocal scanner mounted on a Nikon TE2000-E microscope with a x60/1.4 NA oil-immersion objective lens and 1.515 immersion oil at room temperature. Images were acquired using Metamorph software (Molecular Devices) driving an electron-multiplying charge-coupled device (CCD) camera (ImageEM, Hamamatsu). At least 10 Z sections (0.2 μm apart) were acquired to produce a stack that was then imported into AUTOQUANT X2 (Media Cybernetics) for deconvolution (10 iterations). Single-channel z-plane images and maximum projections were generated in METAMORPH, with overlays performed in Fiji (ImageJ), including adjustments to brightness and contrast.

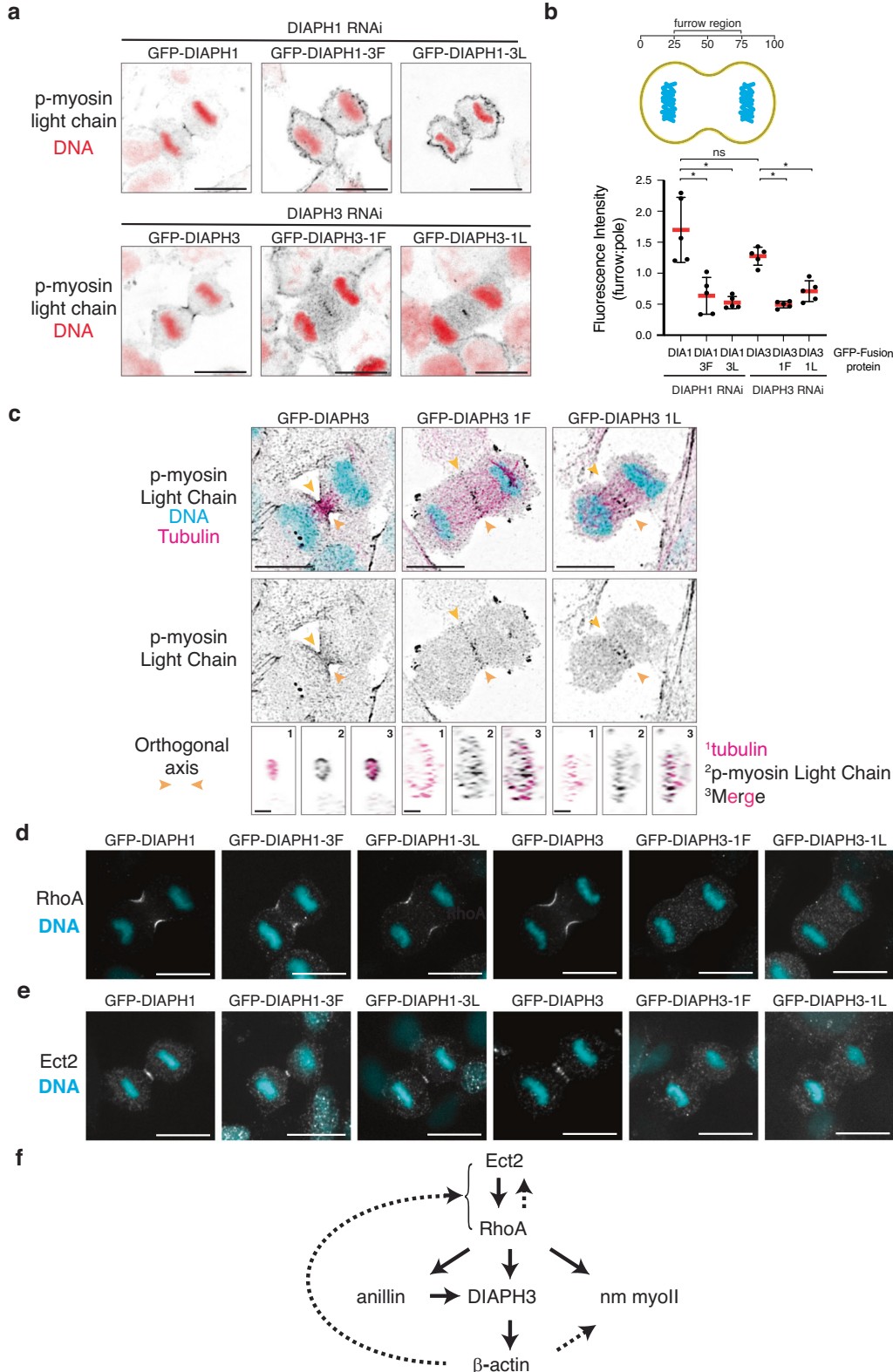

## In cellula mitochondrial localization assay

HeLa cells were cultured in Dulbecco's Modified Eagle Medium (DMEM; Sigma) with 10% heat-inactivated fetal bovine serum (FBS; Invitrogen) and 1% Penicillin-Streptomycin (Invitrogen) and maintained in a 37 °C incubator with 5% $CO_2$. For transient transfection, plasmids expressing Tom20-GFP-DIAPH CT chimeras (see Supplementary Table 1) were transfected into HeLa cells grown on glass coverslips with Lipofectamine 2000 (Invitrogen). For one well of a six-

well plate, 2.5 μg of DNA and 7.5 μL of Lipofectamine 2000 were mixed into 250 μL OptiMEM media (Sigma). The mixture was incubated at room temperature for 20 min before addition to cells. After 24 h of incubation, cells were fixed as described above.

Quantification of colocalization analyses between GFP-Tom20-tagged DIAPH1, DIAPH3, DIAPH1-3F, DIAPH3-1F, DIAPH1-3L, DIAPH3-1L with β- or γ-actin were performed using the Coloc2 plugin of ImageJ (Fiji package, v 2.0.0-rc-69). A total of 20 sets of images in each co-

**Fig. 7 | β but not γ-actin at the furrow supports nm myoII, RhoA, and Ect2 localization at the furrow. a** Micrographs of HeLa cells expressing DIAPH1 and 3 linker mutants in the absence of either endogenous DIAPH1 or DIAPH3, fixed and stained for phospho-myosin light chain (black) and DNA (red). Scale bars represent 10 μm. The micrographs presented are representative of three independent experiments. **b** Top: Cartoon illustrating the region of phospho-myosin light chain fluorescence intensity measurement (yellow). Bottom: Quantitation of the fluorescence intensity of phospho-myosin staining within the furrow and polar regions of anaphase cells described in panel **a**. $n = 5$ cells analyzed across three independent experiments. Data were presented as mean (red bars) ± SD. "ns" = 0.42, *$p = 7.93 \times 10^{-3}$ as determined by two-sided Mann–Whitney non-parametric tests. **c** Micrographs of HeLa cells depleted of endogenous DIAPH3 and expressing either DIAPH3, DIAPH3 1 F, or DIAPH3 1 L stained for phospho-myosin light chain (black)

and alpha-tubulin (magenta) to visualize microtubules. Orthogonal views are presented along the green dashed lines in whole cell images. Scale bars represent 10 μM for whole cell images and 2 μM for orthogonal views. The micrographs presented are representative of three independent experiments. **d** Micrographs of HeLa cells expressing DIAPH1 and 3 linker mutants in the absence of either endogenous DIAPH1 or DIAPH3, fixed and stained for RhoA and DNA (blue). Scale bars represent 10 μm. The micrographs presented are representative of three independent experiments. **e** Micrographs of HeLa cells expressing DIAPH1 and 3 linker mutants in the absence of either endogenous DIAPH1 or DIAPH3, fixed and stained for Ect2 and DNA (blue). Scale bars represent 10 μm. The micrographs presented are representative of three independent experiments. **f** Schematic outlining a possible pathway whereby β-actin can feedback to influence RhoA.

staining group were analyzed to calculate the auto-threshold Manders' overlapping coefficient (tM1) of equally sized regions of interest (ROI) at the leading edge of the plasma membrane. Both channels of each co-staining set were first subjected to background subtraction with a rolling ball algorithm (radius = 50.0) in ImageJ. To calculate tM1 at leading-edge ROIs, the 'Freehand' tool in ImageJ was used to draw leading-edge segments of both channels of a co-staining set. The two channels were then analyzed by the Coloc2 plugin using the Costes method threshold regression (Costes, 2004). The point spread function (PSF) was set to 10, with Costes randomization set to 100. The M1 channel was assigned as GFP-Tom20-DIAPH1/DIAPH3/DIAPH1-3F/DIAPH3-1F, while the M2 channel was assigned as β- or γ-actin as indicated in the graph. The values of tM1 (above threshold) were exported for statistical analysis. Control groups were set up to compare a DIAPH1 (or phalloidin) single-channel image with the exact same image (0° rotation, positive control) or rotating it 90° (negative control). Two-sided non-parametric Mann–Whitney tests were used to calculate *p* values in GraphPad Prism (v10.2.0).

## Western blotting analysis

Following electrophoretic transfer, nitrocellulose or PVDF membranes were incubated in 5% skim milk powder (Nestle) in TBS-T for 1 h at room temperature. After washing, membranes were incubated with indicated primary antibody (antibodies used in this study are reported in Supplementary Table 3) before washing and further incubation with horseradish peroxidase-conjugated secondary antibody. Membranes were developed with chemiluminescent solution (Life Technologies) for 5 min at room temperature and visualized using a Bio-Rad MP Imager (Bio-Rad) controlled by ImageLab software (Bio-Rad).

## In vitro actin pelleting assays

To assess if DIAPH1 and DIAPH3 chimeras preferentially produce β- or γ-actin filaments, 1 mg lyophilized platelet actin (85% β-actin: 15% γ-actin; Cytoskeleton Inc.) or gizzard actin (20% β-actin: 80% γ-actin; Cytoskeleton Inc.) was first resuspended in 50 μL $H_2O$, then 150 μL G-buffer (5 mM Tris-HCl pH 8.0, 0.2 mM $CaCl_2$, 0.2 mM ATP, 0.5 mM DTT) was added to make a 58 μM actin stock that was incubated for 2 h on ice. Actin was then further diluted with G-buffer to a final concentration of 2 μM and freshly purified DIAPH1 and three chimeras in 2 mM Tris pH = 8.0, 0.5 mM β-mercaptoethanol was added to the actin solution to a final concentration of 5 nM. The actin polymerization reaction was initiated by adding polymerization buffer (25 mM Tris, pH 7.0, 50 mM KCl, 2 mM $MgCl_2$, 0.1 mM ATP) and the mixture was incubated at 23 °C for 5 min. Actin filaments were recovered by ultracentrifugation at 157,000×*g* for 20 min in a TLA 120.2 rotor-equipped Beckman TL-100 Ultracentrifuge (Beckman Coulter). Supernatants were carefully separated from pellets, and both were boiled with SDS sample buffer and analyzed by SDS-PAGE and Coomassie blue staining or by immunoblot using anti β- or γ-actin antibodies (see Supplementary Table 3). To determine the comparative amount of actin in soluble unpolymerized fractions (S) compared to the insoluble polymerized fractions (P), the

Coomassie-stained gels were scanned in a Bio-Rad MP Imager (Bio-Rad) and the band intensities measured using ImageLab software (Bio-Rad). The intensity of the actin band in the platelet actin alone pelleting assay that remained in solution (soluble unpolymerized fraction) was defined as the control, and this intensity was set to 1.0. The intensity of bands in the different fractions and reaction conditions were then compared to the control. Each actin pelleting assay was repeated at least three times.

## In vitro actin on-glass polymerization assays

To visualize the production of actin filaments on glass, polymerization reactions were performed on 18 mm diameter glass coverslips pre-treated with sterile poly-lysine solution (0.5 mg/mL in 0.15 M borate buffer, pH = 8.3) for 6 h followed by washing in sterile $H_2O$ and air-drying for another 2–4 h prior to use. Polymerization assays were set up as described above, but immediately after the initialization of polymerization, the reaction mixtures were transferred to poly-lysine-coated coverslips and incubated at room temperature for 20 min. The solution was then carefully removed from the coverslips before they were directly fixed with formaldehyde, post-fixed with methanol, and stained with anti β- or γ-actin antibodies as described above. The number of filament ends per field of view was then counted as a measure of the number of filaments.

## Generation of stable cell lines

HeLa cell lines inducibly expressing GFP-DIAPH1 and GFP-DIAPH3 were described previously[13,14]. cDNAs encoding GFP-DIAPH1-3F, GFP-DIAPH3-1F, GFP-DIAPH1-3L, and GFP-DIAPH3-1L were cloned into a modified pcDNA5 FRT/TO plasmid downstream and in frame with an inserted GFP, using the In-Fusion Cloning Kit (Takara Bio USA, Inc.). Stable HeLa cell lines inducibly expressing these cDNAs under the control of the Tet repressor were generated using the Flp-In system (Life Technologies) using HeLa cells that contained a single FRT site (HeLa FRT/TO cells, a gift of Dr. Laurence Pelletier) (Renshaw et al. 2014). The resulting cell lines were cultured in DMEM (Sigma) supplemented with 10% fetal bovine serum (Life Technologies), 5 μg/mL blasticidin (Bioshop), and 200 μg/mL hygromycin (Bioshop) in a 5% $CO_2$ atmosphere at 37 °C. GFP fusion protein expression was induced by incubating stable HeLa cell lines with 1 μg/mL of doxycycline for 24 h and expression of the GFP fusion protein detected by Western blotting with an anti-DIAPH1 or an anti-DIAPH3 antibody (Supplementary Table 3).

## siRNA treatment and multinucleate assays

HeLa cells were transfected with 40 nM double-stranded DIAPH1 or DIAPH3 siRNA directed against the 3′UTR of the endogenous transcript (Supplementary Table 2) using Lipofectamine 2000 (Invitrogen) following the manufacturer's instructions. For rescue experiments, 16–24 h after siRNA treatment, cells were treated with 1 μg/mL doxycycline (Bioshop) to induce the expression of GFP-DIAPH transgenes that are siRNA resistant as the transcripts generated possess a bovine growth hormone 3′UTR rather than the endogenous 3′UTR. All siRNAs

used were obtained from Integrated DNA Technologies (IDT). For multinucleate assays, cells were scored for the presence of either one or multiple nuclei as indicated by Hoechst staining.

## Fluorescence intensity measurements

To measure actin and phospho-myosin light chain fluorescence intensities, single z-plane images of anaphase cells treated, fixed, and stained, as indicated in corresponding figures, were divided into polar and furrow regions in Fiji as described below. Duplicate images were converted to binary masks and the "Wand" tool was used to generate an ROI outlining the whole cell. This ROI was applied to non-masked images and pixel intensities within polar and furrow regions were summed and used to calculate a fluorescence intensity ratio. For phospho-myosin light chain intensity measurements, a 0.05-μm thick curved line centered on the perimeter of the cell boundary mask was utilized to generate a plot profile of pixel intensities along its length. The summed pixel intensities along segments of the line corresponding to furrow and polar regions were used to calculate a furrow:pole fluorescence intensity ratios. Two-sided non-parametric Mann–Whitney tests were performed to calculate $p$ values in GraphPad Prism (v.10.0.2).

## Live imaging, ingression, and cell elongation rate assays

Cells treated as indicated cultured on circular glass coverslips, thickness no.1, diameter 25 mm (Fisher Scientific) were mounted into a heated chamber with a 5% $CO_2$ atmosphere at 37 °C (Live Cell Instrument Systems) in dyeless DMEM media (Thermo Fisher) supplemented with 10% FBS (Invitrogen). The chamber was in turn mounted on the Nikon TE2000-E microscope described previously equipped with a PlanFluor 20X/0.45 NA Ph1 objective (Nikon). Timelapse video microscopy was used to follow cells with a stack of images (z-steps of 1 μm) taken at regular 1 or 3-min intervals captured on an ORCA-ER CCD camera (Hamamatsu). Furrow ingression and cell elongation measurements were performed on a single, central z-slice for each timepoint using the "Straight Line" function in Fiji. The distance between the furthest points of the two poles of a dividing cell was used for cell elongation measurements, while the distance between the two closest points on either side of the ingressing cytokinetic furrow was used for ingression measurements. Measurements were imported into GraphPad Prism (v.10.0.2) and simple linear regressions were performed to determine ingression and elongation rates for each cell. For ingression rate assays, $n = 5$ cells per condition, except for "DIA1 Control siRNA -dox" and "DIA3 Control siRNA -dox" where $n = 10$ and 6, respectively. Analyzed cells were collected across three independent experiments, except for "DIA1 Control siRNA -dox" and "DIA3 Control siRNA -dox" where cells were analyzed across 6 and 5 independent experiments, respectively. For elongation rate assays, $n = 5$ cells per condition,

except for "DIA1 Control siRNA -dox", "DIA1-3L Control siRNA -dox", and "DIA3-1L Control siRNA -dox" where $n = 10$, 6, and 7, respectively. Analyzed cells were collected across at least three independent experiments, except for "DIA1 Control siRNA -dox", "DIA1-3L Control siRNA -dox", and "DIA3-1L Control siRNA -dox" where cells were collected across 6, 5, and 5, independent experiments respectively. $P$ values were calculated by two-sided non-parametric Mann–Whitney tests using GraphPad Prism software (v.10.0.2).

## Ingression and furrow instability indices

For all treatment conditions, cells grown on glass coverslips were formaldehyde-fixed and stained with rhodamine-phalloidin and Hoechst as previously described. Only Anaphase B cells were considered for analysis; these were determined by the combined presence of condensed chromosome staining via Hoechst and at least some degree of cytokinetic furrow ingression by phalloidin staining. Single Z-slice images of cells meeting these criteria were acquired on a Nikon Eclipse E800 epifluorescence microscope fitted with an ORCA-ER CCD camera (Hamamatsu) controlled by Metamorph software (v.7.6.5.0).

As pictured in Fig. 5c, the distance of the furrow's indentation was measured on both of its sides in relation to the same nascent daughter cell (i.e., relative to the same half of the dividing cell) using the "Straight Line" function in Fiji. For each cell, the greater of these two distances was used as the divisor to calculate the Ingression Index, such that all ratios ≥1. $N = 20$ dividing cells measured for each condition, derived from at least three biological replicates. The same cells analyzed for Ingression Index measurements were also analyzed for Furrow Instability measurements. As pictured in Fig. 5c, the distances from each pole of the dividing cell to the center of its furrow were measured with the "Straight Line" function in Fiji. For each cell, the greater of these two distances was used as the divisor to calculate the Furrow Instability Index, such that all ratios ≥1.

## Furrow and polar bleb quantification

Cells grown on glass coverslips were formaldehyde-fixed and stained with rhodamine-phalloidin and Hoechst as previously described. Only Anaphase B cells were selected for analysis. Single Z-slice images of cells were acquired on the Nikon Eclipse E800 epifluorescence microscope described previously. Images were imported into Fiji, where the "Straight Line" function was used to measure the length of each half of the dividing cell (from furrow to pole) as revealed by phalloidin staining. Each half-cell was then divided into two regions of equal length, with the furrow-adjacent region classified as "furrow" and the polar cortex-adjacent region classified as "polar". The number of blebs were counted in each of these regions and pooled with the counts from the corresponding regions from the other half of the same cell. In some cases, precise measurement of 'polar' blebs was impeded by the polar regions being out-of-focus with the furrow region of a given cell; accordingly, measurements for "polar" blebs in these cells were discarded. $n ≥ 10$ cells scored for furrow bleb quantification and $n ≥ 7$ cells scored for polar bleb quantification across at least three independent experiments.

## Model building

The model of FMNL3 and actin was built in PyMol from coordinates deposited in PDB (code 4EAH[37],). Models for the DIAPH3 FH2 domain:β-actin complex were predicted by Alphafold2[40] in multimer mode[39]. The sequences of human β-actin (full length) and the residues corresponding to the FH2 domain of human DIAPH3 (residues 636 to 1034) were subjected to model building on Alphafold2 installed on Google collaboratory. In all five of the Alphafold models generated, the β-actin N-terminus and the FH2 linker region were of low confidence, these areas overlap almost perfectly with the regions not seen in the FMNL3:actin crystal structure.

The low confidence regions of the Alphafold2 model, specifically, the β-actin N-terminus ($^1$DDDIA$^5$), and DIAPH3 FH2 domain ($^{687}$CCQQKERREEEDIEEKKSIKKKIK$^{710}$), were removed from the top ranked Alphafold PDB and the resulting file was used as an input in the IPDConformerGenerator software package[42]. We simultaneously generated 10 conformers each of the β-actin N-terminus and the DIAPH3 FH2 domain. IPDConformerGenerator was run using the local disordered region sampling (LRDS) module using the default settings and with ANY ("--dany") flagged for secondary structure sampling[41]. As a result, the secondary structure sampling arises from the intrinsic structural propensities of the input sequence fragments as found in the PDB.

## Quantification and statistical analysis

The quantification of the band intensities in the western blots was performed using ImageLab software (Bio-Rad). Where noted, non-parametric, two-sided Mann–Whitney tests were performed to calculate $p$ values in GraphPad Prism (v10.2.0).

## Reporting summary

Further information on research design is available in the Nature Portfolio Reporting Summary linked to this article.

## Data availability

Any plasmids and cell lines generated during and/or analyzed in this study are available from the corresponding author upon request. Source data are provided with this paper.

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

## Acknowledgements

We would like to thank Antonina Mazur for the insightful discussion and Chen Tuo for generating images of previously published actin-formin complexes. T.F.M. has infrastructure support provided by the Canadian Foundation for Innovation and operating support from the Natural Sciences and Engineering Research Council of Canada (RGPIN-2018-06546). C.A.M. is supported by a Canadian Institute for Health Research Project grant (MOP-512789). A.W. is supported by the Canadian Institute of Health Research (MOP-511834) and the National Science and Engineering Research Council (RGPIN-2019-05782).

## Author contributions

R.S. performed all biochemistry, except for western blots in Supp Fig. 3c, d that were performed by T.C.P. R.S. performed all cell biological assays, except for those described in Figs. 5d–i and 7b, c that were performed by T.C.P. G.B.C. and T.F.M. performed computer modeling of formin-actin complexes. R.S., T.C.P., B.D.L., C.A.M., and A.W. conceived experiments. The manuscript was written by R.S., T.C.P., B.D.L., and A.W., and was edited by all authors. Funding was provided by A.W., C.A.M., and T.F.M.

## Competing interests

The authors declare no competing interests.
