## [Peer Review File · Nature Communications]

The DIAPH3 linker specifies a β -actin network that maintains RhoA and Myosin-II at the cytokinetic furrowREVIEWER COMMENTS

Reviewer #1 (Remarks to the Author):

Shah et al. in this manuscript described β -actin is indispensable for cytokinesis and required for maintaining key proteins such as RhoA and myosin-II at the cleavage furrow. They confirm the specificity of the formins DIAPH1 and DIAPH3 to polymerize distinct actin subunits. In addition, in a series of in vitro and in vivo experiments, the authors identified the linker regions of DIAPH3 FH2 confers its β -actin isoform specificity. Contribution of β and γ -actin towards the distinct cellular process including furrow ingression is presented very well. They showed the two actin subunits cannot be substituted with one another for their function. This is a solid story with rigorous data typical of the Wilde lab. However, the discoveries in this story are incremental and expected compared to the groundbreaking findings in their Nat Comm story in 2017. Concerns and comments listed below should be addressed to improve the manuscript.

Major concerns

1. Title: the majority of the data are related to the formins DIAPH1 and DIAPH3 for their functionalities and specificity towards actin subunits, which I feel that it is justified to mention formin DIAPH3 in the title.
2. Although the experiments on mistargeting β and γ -actin are solid and the conclusion reasonable, it cannot be ruled out the cytokinesis defects are caused by altering of actin and formin levels at cell cortex and cleavage furrow. Can the relevant constructs in stable cell lines expressed under the native promoters? If not possible, the protein intensity at the furrow should be measured, and overall protein expression levels should be tested by Western blottings. For some proteins, the sequences near the NH2 termini are also important for expression levels. So testing the protein levels will also confirm the importance of NH2 termini of DIAPH1 and DIAPH3 in actin specificity.
3. Authors showed Tom20-GFP DIAPH3-1F and Tom20-GFP DIAPH3-1L can nucleate both β and γ -actin in Figures 1 and 3 in colocalization and in-vitro studies. Then they showed GFP-DIAPH3-1L can only nucleate γ -actin in Figure 4. Can the authors explain what the reason was. Both studies were done in-cellula so the post-translational modifications cannot be the reason. Maybe their construct is not working properly in the later experiments when they showed γ -actin specific nucleation by DIAPH1? Did they use the same construct just without Tom20 in Figure 4? In this manuscript, more experiments are supporting DIAPH1 as nucleator of both β and γ instead for just β .
4. In Figure 7, Myo2 localized to furrow and spread to cortex when DIAPH1 is depleted along with the expression of GFP-DIAPH1 and GFP-DIAPH1-3F respectively, which indicates β -actin dictates the Myo2 localization. But when β -actin is depleted along with substitution of β to γ -actin using GFP-DIAPH3-1F or GFP-DIAPH3-1L, why Myo2 is still localized to furrow with same furrow/pole intensity as GFP-DIAPH3? Maybe DIAPH1 is nucleating enough β -actin to localize Myo2 on furrow? It's also likely myosin-II localization is largely independent of actin as shown in many systems. So it's not convincing to say that β -actin dictates myosin-II localization. The complications should be thoroughly discussed.

Minor comments

1. There is no Supp Figure 5 (mentioned on page 16 paragraph 3) and Supp Table 4 (mentioned on page 22 paragraph 2) provided with the manuscript.
2. How many Z sections were captured with 0.2 μ m spacing for immunofluorescence and microscopy analysis?
3. Red and green combinations are not favorable for colorblind readers, you can choose other combinations i.e., blue and red or green and magenta.
4. Page 14 paragraph 2 line 6,7, you compared cell elongation rates between GFP-DIAPH1 expressed and unexpressed with depletion of endogenous DIAPH1 and found rate is rescued/increased to 0.472 but not by expression of GFP-DIAPH1-3L. Here you mentioned elongation rate 0.348 with $p=0.0303$, but I think this p value in figure 6D is mentioned for

comparison between GFP-DIAPH1-3L under unexpressed conditions with or without endogenous DIAPH1. In addition, the p values shown in the figures are crowded, and rounded up to the second decimal is enough.

Minor corrections:

1. Page 3 paragraph 2 line no 3, change "restricted tissue" to "restricted to tissue" on.
2. Page 8 paragraph 1 line no 8, should include supp "Fig 1 C-E".
3. Page 12 paragraph 1 line no 11, "Fig 5D, E" should be only "Fig 5D".
4. Page 12 paragraph 2 line no 10, add Fig 5G.
5. Page 13 paragraph 2 line no 8, I think 6B figure is not involved so you can just write "Fig 6 and C".
6. Page 14 paragraph 2 line no 5, write exact p value.
7. Page 15 paragraph 2 line 9, RhoA and Ect2 are represented in both Figure 7C and D.
8. Page 17 paragraph 1 line no 4, change "apposition" to "a position".
9. Page 20 paragraph 1 line 1, change "pGEX-6P-2-DIAPH1-CTand" to "pGEX-6P-2-DIAPH1-C and".

Figure legends

1. Please specify amino acids ranges for CT in Figure 1A.
2. Figure 5B: change "GFP-DIAPH1 or GFP-DIIAPH1" to "GFP-DIAPH1 or GFP-DIAPH1".
3. Figure 5E: change "presence of absence" to "presence or absence".
4. Figure 7C and D, DIAPH1 and DIAPH3 RNAi are used. Authors should mark which RNAi is used in which picture.

Reviewer #2 (Remarks to the Author):

Reviewer #3 (Remarks to the Author):

This paper is an extension of very interesting work by Chen (2017 and 2021), who presented data consistent with DIAPH3 in the cleavage furrow promoting polymerization of beta-actin but not gamma-actin and DIAPH1 promoting the polymerization of gamma-actin throughout the rest of the cortex. This manuscript has evidence that the linker peptides of the two formins provide the specificity for the two actin isoforms. Using chimeric constructs with linkers exchanged between the two formins, the authors reversed the distributions of the two actin isoforms and demonstrated that they cannot substitute for each other during cytokinesis. Furthermore, beta-actin in the cleavage furrow is required to localize myosin-II and RhoA.

The topic is interesting and important for understanding cytokinesis and the specialized contributions of formin isoforms. The authors point out that the mechanisms explaining these dramatic observations are still mysterious, so this work should motivate additional research.

I am enthusiastic about publication of the work despite questions about the mechanisms that extend back to previous papers from this group. At the end of the review, a note to the authors explains problems with the design and interpretation of their actin polymerization experiments and suggests better approaches.

Historical context

- From an evolutionary point of view, it might be worth mentioning the precedent for specific formins assembling the contractile ring (Cdc12) and actin cables (For3) in fission yeast, which

diverged from animals more than 1 million years ago.

- How does the present work on DIAPH3 relate to the work by Watanabe and colleagues, who showed that mDia2 is important for cytokinesis in mouse cells (especially RBC precursors) but that mice with deletions of mDia1 or mDia3 or both “develop to adulthood?”

Presentation

The paper is clear and easy to read, although it would be even better, if

- the text was less conversational (introductory background material in many paragraphs on results is not necessary) and
- long paragraphs on multiple topics were broken into separate paragraphs with strong topic sentences. To note just one example on page 6, start a new paragraph with “To determine if the FH2 domain dictates...”

Detailed comments

Page 6, Fig 1: Why does DIAPH3-1F assemble both beta- and gamma-actin on mitochondria, when previous work emphasized that DIAPH1 specifically assembles gamma-actin?

Page 7: The text should explain that “constitutively active DIAPH1, DIAPH3, DIAPH1-3F and DIAPH3-1F formins” are the C-terminal fragments. The legend for Fig 2 should give the incubation times.

Page 8, Fig S1C/D: “all formins generated homopolymers” is an amazing observation that is highly unexpected given the ability of actins from very divergent sources to copolymerize. It suggests that an actin monomer in solution can recognize the identity of the two actin subunits on the barbed end of the growing filament. How can you explain this? I have no idea, so the authors should acknowledge that this observation is unexpected and unexplained. Might the problem be with the methods? Are the observations in Fig S1C/D reliable? For example, why is the gamma-actin fluorescence so weak in the sample with gizzard actin alone?

Page 9, Fig 3: the FMNL3/actin crystal structure of Thompson et al. without contacts between the two actins is unlikely to be the form in solution. The authors should consider the MD simulations that have been done to refine cocrystal structures of other FH2/actin cocrystals. Did the authors attempt to make homology models of DIAPH1 and 3? They might be more revealing than the FMNL3 structure. Did you consider that the length, rather than the sequence of these two linkers might specify actin isoforms? Mitochondrial localization and copelleting with C-terminal fragment chimeras support the conclusion that the linker peptides are responsible for the actin isoform specificity of DIAPH1 and 3 but measuring nucleation and elongation rates for the two actin isoforms would be stronger support. (See the note on methods at the end.)

Page 9, Fig 4: Full length formins with linker swaps were expressed in Hela cells with endogenous wild type formins. Then siRNA was used to deplete a wild type formin. Explain how this avoided depletion of the linker swapped formin. How effective is the siRNA depletion of the formins? In panel B, top row, why is the strong staining of beta-actin in dox- missing in the cell with dox+?

The quality of the blots in Fig S3 does not support the statement “expression of GFP fusion proteins at near endogenous levels (Supp Fig 3).” This expression at lower than wild type levels may have influenced the outcome of experiments where the wild type proteins were depleted. This potential artifact should be addressed in the text. Was expression similar in all the cells with dox? If not homogeneous, interpreting the data is complicated.

Fig. 5: the redistribution of the actin isoforms depending on the linker is impressive.

Page 11: Do you really mean “ $57.8 \pm 2.47\%$ multi-nucleate cells in the GFP-DIAPH3 cell line compared to 4.82 ± 0.840 for control siRNA treated cells?”

Page 13: The small effect of depleting DIAPH3 on furrow ingression rate is surprising. Do the authors have any explanations? Divide the last paragraph into two parts.

Page 14, Fig 7A: It would be desirable to have better documentation (more examples, 3D reconstructions and perhaps colocalization with microtubules) to verify the important finding that "upon replacing beta-actin with gamma-actin at the furrow by expressing GFP-DIAPH3-1L in cells depleted of endogenous DIAPH3, phospho-myosin light chain was absent from the furrow membrane and instead concentrated at the spindle midzone." In Fig 7B specify on the figure that the staining is for phosphorylated myosin-II. What mechanisms might explain the redistribution? Do beta- and gamma-actin have different affinities for myosin-II or any actin binding protein that might influence the localizations of myosin-II and RhoA?

Page 16: Fig S5 was not included.

Page 17: The text states "there may be further complexity in the specificity process for some formins as in vitro the constitutively active DIAPH1 polymerized both beta- and gamma-actin homopolymeric filaments while in cellula full length DIAPH1 at the cell cortex produces only gamma-actin filaments." The obvious difference is the presence of profilin in cells, which influences both nucleation and elongation by formins.

Minor concerns

Given the small numbers of cells, are 3 significant figures (for example "ingression index of 1.11 ± 0.0601 , 0.780 ± 0.0861 " etc.) justified for most of measurements?

Page 5 and elsewhere: Myosin is not a proper noun.
Throughout: Is "in cellula" a real term?

Page 20, Table 3 and elsewhere. Paraformaldehyde is a solid which is converted formaldehyde for use as a fixative. Dilution factors are meaningless without knowing the stock concentration of the antibodies.

Critique of the methods used to characterize actin polymerization:

The overall conclusions in Chen 2017 and 2021 and the current paper may be correct, but the methods used to characterize the formins make it difficult to interpret the data. In particular, none of the data actually show that DIAPH3 or DIAPH1 selectively stimulate nucleation of actin isoforms. The data do not rule out alternative hypotheses such as both formins nucleating equally well but then elongating filaments at much different rates. Therefore, using "assembly" rather than "nucleation" would safer.

Although doing proper experiments to characterize DIAPH1 and DIAPH3 is beyond the scope of this study, the authors should be aware of and acknowledge the following issues:

(1) Neither Chen paper (2017 and 2021) uses state-of-the-art quantitative assays for nucleation or elongation of actin filaments. During spontaneous polymerization assays such as Chen (2017) Figs 2 and 3b, both the rate of nucleation and the rate of elongation contribute to the accumulation of polymerized actin (measured by pyrenyl-actin fluorescence) over time. Elongation rates are measured most simply by TIRF microscopy. Knowing the elongation rate, one can calculate the concentration of growing filament ends from the slope at every point along the spontaneous polymerization time course. The change in the concentration of growing ends vs. time gives the nucleation rate. One can also estimate crudely the nucleation rate by observing the number of filaments by fluorescence microscopy (Chen 2017, Fig 5C). However, the samples in that experiment were taken after 20 min, very late in the polymerization process near steady state. Earlier time points would have been more informative. Furthermore, the method to measure "Filaments per field" is not explained. Is this filament ends or total polymer?

(2) Chen (2017) polymerized Ca-ATP-actin by adding KCl and MgCl₂, conditions where the slow

exchange of Ca for Mg by actin monomers has been known to be rate limiting for nucleation for almost 40 years. To avoid this unphysiological divalent cation exchange reaction, virtually all spontaneous polymerization experiments are done by first exchanging Ca for Mg on actin monomers and then adding KCl, EGTA and MgCl₂ to assemble Mg-ATP-actin monomers.

(3) Formins slow elongation, so one must know elongation rates to interpret the time course of spontaneous polymerization or measure nucleation rates (Chen 2017, Figs 2 and 3b). The data in those figures is certainly consistent with the CT-fragment stimulating nucleation but not definitive proof.

(4) All the polymerization assays are done without profilin, which is bound to actin monomers in cells. Profilin inhibits nucleation by formins more strongly than nucleation by pure actin monomers but increases the rate of elongation by formin constructs with FH1 domains such as the CT fragments in these papers. Therefore, the characterization of the two formins should have included polymerization with and without profilin.

Response to Reviewers

We would like to thank the reviewers for their positive and detailed analysis of our manuscript. Based on their comments and in accordance with their suggestions we have revised the manuscript by including further data, the building of molecular models, additional descriptions and further discussion. A point-by-point response to each reviewer comment is included below after each reviewer query.

Reviewer #1 (Remarks to the Author):

Shah et al. in this manuscript described β -actin is indispensable for cytokinesis and required for maintaining key proteins such as RhoA and myosin-II at the cleavage furrow. They confirm the specificity of the formins DIAPH1 and DIAPH3 to polymerize distinct actin subunits. In addition, in a series of in vitro and in vivo experiments, the authors identified the linker regions of DIAPH3 FH2 confers its β -actin isoform specificity. Contribution of β and γ -actin towards the distinct cellular process including furrow ingression is presented very well. They showed the two actin subunits cannot be substituted with one another for their function. This is a solid story with rigorous data typical of the Wilde lab. However, the discoveries in this story are incremental and expected compared to the ground-breaking findings in their Nat Comm story in 2017. Concerns and comments listed below should be addressed to improve the manuscript.

Major concerns

1. Title: the majority of the data are related to the formins DIAPH1 and DIAPH3 for their functionalities and specificity towards actin subunits, which I feel that it is justified to mention formin DIAPH3 in the title.

Response: The title of the manuscript has been modified to include DIAPH3.

2. Although the experiments on mistargeting β and γ -actin are solid and the conclusion reasonable, it cannot be ruled out the cytokinesis defects are caused by altering of actin and formin levels at cell cortex and cleavage furrow. Can the relevant constructs in stable cell lines expressed under the native promoters? If not possible, the protein intensity at the furrow should be measured, and overall protein expression levels should be tested by Western blottings. For some proteins, the sequences near the NH2 termini are also important for expression levels. So testing the protein levels will also confirm the importance of NH2 termini of DIAPH1 and DIAPH3 in actin specificity.

Response: We agree that the levels of expression of proteins is critical. Whilst we have not been able to successfully utilize endogenous promoters, we were able to use the Tet-inducible system to express proteins at near endogenous levels. We have included further blots to demonstrate this in Supp Fig 3C,D. An original goal of the study was to “invert” the cytokinetic localization of the β and γ -actin networks. However, as swapping the linker regions between DIAPH1 and 3 led to division defects, we feared expressing these mutants under the control of endogenous promoters would produce an aberrant, multinucleated cell population whose mitotic phenotypes

could be hard to parse between disrupted formin function and their increased aneuploidy. The Tet-inducible system allows for expression of the deleterious mutant constructs in otherwise normal HeLa cells for a small number of division cycles before analysis, bypassing this problem. However, we have not been successful in expressing both mutants in a single cell.

3. Authors showed Tom20-GFP DIAPH3-1F and Tom20-GFP DIAPH3-1L can nucleate both β and γ -actin in Figures 1 and 3 in colocalization and in-vitro studies. Then they showed GFP-DIAPH3-1L can only nucleate γ -actin in Figure 4. Can the authors explain what the reason was. Both studies were done in-cellula so the post-translational modifications cannot be the reason. Maybe their construct is not working properly in the later experiments when they showed γ -actin specific nucleation by DIAPH1? Did they use the same construct just without Tom20 in Figure 4? In this manuscript, more experiments are supporting DIAPH1 as nucleator of both β and γ instead for just β .

Response: In Figures 1 and 3 only the C-terminal half of DIAPH1 (amino acids 583 to 1272) that includes the FH1, FH2 and DAD domains was fused to Tom20-GFP and expressed in cells. However, in Figure 4 full-length DIAPH1 (wildtype or mutant) were fused downstream of GFP and expressed in cells. We have previously shown (Chen et al 2021) and report again in this study (Fig 5B) that the GFP tagged wildtype construct can rescue siRNA-mediated DIAPH1 depletion phenotypes suggesting it is fully functional. We do not yet know the mechanism through which full-length DIAPH1 only generates γ -actin polymer when correctly targeted within a dividing cell while the C-terminal fragment targeted to the mitochondria generates both β and γ -actin polymer. As the reviewer suggests it is unlikely to be through a post-translational modification Alternative models could include other domains of DIAPH1 contributing to specificity or that some of DIAPH1's perceived specificity is due to polymerization kinetics of different actin isoforms rather than an absolute actin isoform preference as maybe the case with DIAPH3. We have expanded the discussion section of the manuscript surrounding this point. While we are actively pursuing this question by first generating new robust tools, it is beyond the scope of the current study.

4. In Figure 7, Myo2 localized to furrow and spread to cortex when DIAPH1 is depleted along with the expression of GFP-DIAPH1 and GFP-DIAPH1-3F respectively, which indicates β -actin dictates the Myo2 localization. But when β -actin is depleted along with substitution of β to γ -actin using GFP-DIAPH3-1F or GFP-DIAPH3-1L, why Myo2 is still localized to furrow with same furrow/pole intensity as GFP-DIAPH3? Maybe DIAPH1 is nucleating enough β -actin to localize Myo2 on furrow? It's also likely myosin-II localization is largely independent of actin as shown in many systems. So it's not convincing to say that β -actin dictates myosin-II localization. The complications should be thoroughly discussed.

Response: The reviewer makes an astute observation and was misled by an error in our quantitation We thank the reviewer for catching this and apologize for our error, which has now been corrected. The micrographs show that in wildtype cells, when β -actin is enriched at the furrow, phospho-myosin II light chain (pMLC) is restricted to the furrow. However, upon the generation of β -actin throughout the cortex of a dividing cell, pMLC now relocates to the cortex

of the dividing cells mimicking the localization of β -actin. In cells where β -actin is depleted from the furrow, the micrographs show that pMLC is no longer localized to the plasma membrane of the furrow rather it now appears on the microtubules of the spindle midzone. We have included further micrographs to demonstrate this relocalization to the spindle midzone (Fig 7C). The confusion stems from our error in the quantitation of the micrographs. In the initial submission, each whole cell was broken into 4 equal sections, the two central ones we termed the equatorial region, the 2 at the extremes the polar region. As that quantitation strategy did not specifically focus on pMLC on the membrane the quantitation revealed that pMLC remained in the equatorial region even though its localization had changed from being on the membrane of the furrow to the spindle midzone microtubules. We apologize for this confusion. We have now re-quantified the images using the same portioning method but now quantifying only the pMLC signal along the plasma membrane of dividing cells. This now reflects the relocalization of membrane-associated pMLC observed in the micrographs and is consistent with our statement that β -actin localization dictates nm-myosinII membrane localization.

Minor comments

1. *There is no Supp Figure 5 (mentioned on page 16 paragraph 3) and Supp Table 4 (mentioned on page 22 paragraph 2) provided with the manuscript.*

Response: We apologize for the omission and have included them in the re-submission.

2. *How many Z sections were captured with 0.2 μ m spacing for immunofluorescence and microscopy analysis?*

Response: The methods section has been amended to clarify the number of Z sections acquired.

3. *Red and green combinations are not favorable for colorblind readers, you can choose other combinations i.e., blue and red or green and magenta.*

Response: We have now modified the color schemes of micrographs in this submission.

4. *Page 14 paragraph 2 line 6,7, you compared cell elongation rates between GFP-DIAPH1 expressed and unexpressed with depletion of endogenous DIAPH1 and found rate is rescued/increased to 0.472 but not by expression of GFP-DIAPH1-3L. Here you mentioned elongation rate 0.348 with $p=0.0303$, but I think this p value in figure 6D is mentioned for comparison between GFP-DIAPH1-3L under unexpressed conditions with or without endogenous DIAPH1. In addition, the p values shown in the figures are crowded, and rounded up to the second decimal is enough.*

Response: We have corrected the text, which now compares the rates of cell elongation between cells depleted of DIAPH1 with and without GFP-DIAPH1-3L expression, in line with the

comparisons made on Figure 6D. We have moved p values from figure panels to figure legends and decreased the number of significant digits reported.

Minor corrections:

1. Page 3 paragraph 2 line no 3, change “restricted tissue” to “restricted to tissue” on.
2. Page 8 paragraph 1 line no 8, should include supp “Fig 1 C-E”.
3. Page 12 paragraph 1 line no 11, “Fig 5D, E” should be only “Fig 5D”.
4. Page 12 paragraph 2 line no 10, add Fig 5G.
5. Page 13 paragraph 2 line no 8, I think 6B figure is not involved so you can just write “Fig 6 and C”.
6. Page 14 paragraph 2 line no 5, write exact p value.
7. Page 15 paragraph 2 line 9, RhoA and Ect2 are represented in both Figure 7C and D.
8. Page 17 paragraph 1 line no 4, change “apposition” to “a position”.
9. Page 20 paragraph 1 line 1, change “pGEX-6P-2-DIAPH1-CTand” to “pGEX-6P-2-DIAPH1-C and”.

Response: The suggested minor modifications have all been implemented.

Figure legends

1. Please specify amino acids ranges for CT in Figure 1A.

Response: We have added the amino acid ranges into the figure legend.

2. Figure 5B: change “GFP-DIAPH1 or GFP-DIAPH1” to “GFP-DIAPH1 or GFP-DIAPH1”.
3. Figure 5E: change “presence of absence” to “presence or absence”.
4. Figure 7C and D, DIAPH1 and DIAPH3 RNAi are used. Authors should mark which RNAi is used in which picture.

Response: We have made all the suggested modifications outlined in items 2 to 4.

Reviewer #2 (Remarks to the Author):

Reviewer #3 (Remarks to the Author):

This paper is an extension of very interesting work by Chen (2017 and 2021), who presented data consistent with DIAPH3 in the cleavage furrow promoting polymerization of beta-actin but not gamma-actin and DIAPH1 promoting the polymerization of gamma-actin throughout the rest of the cortex. This manuscript has evidence that the linker peptides of the two formins provide

the specificity for the two actin isoforms. Using chimeric constructs with linkers exchanged between the two formins, the authors reversed the distributions of the two actin isoforms and demonstrated that they cannot substitute for each other during cytokinesis. Furthermore, beta-actin in the cleavage furrow is required to localize myosin-II and RhoA.

The topic is interesting and important for understanding cytokinesis and the specialized contributions of formin isoforms. The authors point out that the mechanisms explaining these dramatic observations are still mysterious, so this work should motivate additional research.

I am enthusiastic about publication of the work despite questions about the mechanisms that extend back to previous papers from this group. At the end of the review, a note to the authors explains problems with the design and interpretation of their actin polymerization experiments and suggests better approaches.

Historical context

- From an evolutionary point of view, it might be worth mentioning the precedent for specific formins assembling the contractile ring (Cdc12) and actin cables (For3) in fission yeast, which diverged from animals more than 1 million years ago.*

Response: We have added further context into the discussion section relating to this point. It presents a very interesting evolutionary note in that organisms with only 1 actin gene possess fewer nucleators, such that one could propose a mechanism whereby actin production at different locations can be regulated by different pathways. However, in metazoans, there is an amplification in the number of actin genes and a considerable amplification of nucleators from which one could propose that in addition to a need to differentially regulate the production of actin networks at specific times and places within a cell, these networks are made from distinct actin isoforms. They may serve to produce biochemically divergent platforms within the cell to coordinate specific functions. This may indeed be a suitable subject for a review or comment article in the future.

- How does the present work on DIAPH3 relate to the work by Watanabe and colleagues, who showed that mDia2 is important for cytokinesis in mouse cells (especially RBC precursors) but that mice with deletions of mDia1 or mDia3 or both “develop to adulthood?”*

Response: The reviewer raises an interesting point that is quite difficult to address in any the field: why do the disruptions of certain factors in tissue culture cells leads to dramatic cellular phenotypes while their disruption in whole animals leads to surprisingly modest phenotypes. Obvious parallels in cytokinesis are the dramatic effects of knockouts of some factors involved in the process of abscission in tissue culture cells and the surprising viability of whole animals deleted of the same factors yielding only tissue specific effects. One likely possibility is that a minority of cells successfully compensate (at least in part) for the defect and go on to develop into whole animals--and thus the dramatic defects in tissue culture are not recapitulated in whole animals. The molecular nature of this compensation could include redundancy between family members, changes in expression levels or perhaps more likely that in the whole animal the cells

are surrounded by other cells and extracellular matrix that may confer physical forces on a dividing cell to overcome the lack of factors that normally provide such force internally. In the future it would be of great interest to perhaps study such events in organoids or simple multicellular agglomerations (spheroids) to address such questions if the corresponding imaging technologies are sufficient to monitor the events.

Presentation

The paper is clear and easy to read, although it would be even better, if

- *the text was less conversational (introductory background material in many paragraphs on results is not necessary) and*
- *long paragraphs on multiple topics were broken into separate paragraphs with strong topic sentences. To note just one example on page 6, start a new paragraph with “To determine if the FH2 domain dictates...”*

Response: While modifying the submission we have endeavoured to make such changes.

Detailed comments

Page 6, Fig 1: Why does DIAPH3-1F assemble both beta- and gamma-actin on mitochondria, when previous work emphasized that DIAPH1 specifically assembles gamma-actin?

Response: This related in part to the response to reviewer 1 above. The mitochondrial targeted constructs only express the C-terminal half (FH1, FH2 and DAD) of the formins. While the mitochondrially targeted DIAPH3-1F generates β - and γ -actin on mitochondria so does the mitochondrially targeted DIAPH1 C-terminal fragment, making the observations consistent with each other. It is only in the context of the full-length DIAPH1 in dividing cells where we observe γ -actin production. We apologize for the confusion and have made this explicit in the text.

Page 7: The text should explain that “constitutively active DIAPH1, DIAPH3, DIAPH1-3F and DIAPH3-1F formins” are the C-terminal fragments. The legend for Fig 2 should give the incubation times.

Response: We have made the corresponding changes to enhance clarity.

Page 8, Fig S1C/D: “all formins generated homopolymers” is an amazing observation that is highly unexpected given the ability of actins from very divergent sources to copolymerize. It suggests that an actin monomer in solution can recognize the identity of the two actin subunits on the barbed end of the growing filament. How can you explain this? I have no idea, so the authors should acknowledge that this observation is unexpected and unexplained. Might the

problem be with the methods? Are the observations in Fig SIC/D reliable? For example, why is the gamma-actin fluorescence so weak in the sample with gizzard actin alone?

Response: We agree: the homopolymerization of actin isoforms is a fascinating and unexpected observation, which we first reported and discussed in our 2017 paper. To our knowledge, prior experiments did not examine the polymers produced directly through methods that would have distinguished between actin isoforms in the same polymerization reaction. For example, Manstein and co-workers used a pyrene-labelled actin-based assay and saw an increase in signal which they interpreted as co-polymerization, but no direct observations of the filaments were made. To address this fully, single molecule nucleation and elongation experiments (as you rightfully suggest later) using differentially labelled actin isoforms of very high purity will be needed to address this fundamental question more directly, of which the latter is the greatest challenge and something we are working toward for future experiments. We have added further elements into the discussion accordingly.

Page 9, Fig 3: the FMNL3/actin crystal structure of Thompson et al. without contacts between the two actins is unlikely to be the form in solution. The authors should consider the MD simulations that have been done to refine cocrystal structures of other FH2/actin cocrystals. Did the authors attempt to make homology models of DIAPH1 and 3? They might be more revealing than the FMNL3 structure. Did you consider that the length, rather than the sequence of these two linkers might specify actin isoforms? Mitochondrial localization and copelleting with C-terminal fragment chimeras support the conclusion that the linker peptides are responsible for the actin isoform specificity of DIAPH1 and 3 but measuring nucleation and elongation rates for the two actin isoforms would be stronger support. (See the note on methods at the end.)

Response: We include additional modelling approaches in collaboration with the newly listed co-authors. The *in silico* molecular dynamics study that predicts salt bridge formation between the residues in the variable extreme actin N-termini and the linker region of DIAPH1 are generally supportive of our model and findings. However, we were reluctant to include models from that study as we had noticed that based on the coordinates deposited in PDB, a region of the FH2 domain passes through a loop of the actin protein sequence, an organizational state not consistent with the *in vivo* state. We have included a figure below highlighting this for the reviewers. We have now built further models using docked alpha fold generated structures (Fig 3D and Sup Fig 3) that support our overall model that the actin N-terminus is most likely to interact with the formin FH2 linker region.

Response Figure 1. Model built from PDB deposited co-ordinates by Aydin et al. Arrow marks point where the formin polypeptide chain passes through a loop of the actin polypeptide chain.

We did consider the length of the linker as a determining factor. This was in the omitted supplementary Figure 5 for which we apologize. DIAPH3 across species consistently has a shorter linker than the DIAPH1. This and a conserved difference in charge would appear to be the simplest model to explain the difference in actin isoform preference of the two formins. We have expanded the discussion of this concept in the Discussion section of the revised manuscript.

Page 9, Fig 4: Full length formins with linker swaps were expressed in HeLa cells with endogenous wild type formins. Then siRNA was used to deplete a wild type formin. Explain how this avoided depletion of the linker swapped formin. How effective is the siRNA depletion of the formins? In panel B, top row, why is the strong staining of beta-actin in dox- missing in the cell with dox+?

Response: We routinely use a strategy whereby we design our siRNAs to the 3'UTR of an endogenous transcript. By expressing wildtype and mutant proteins in our plasmid-based system that generates transcripts with a bovine growth hormone 3'UTR, we avoid targeting of transgenes. We have added further descriptions in the text and the methods and have included further blots to confirm that the siRNA used targets endogenous transcripts and not those generated through ectopic expression. In relation to the β -actin staining intensity in Figure 4B, we suspect that the β -actin generated at the cortex by induction of GFP-DIAPH1-3L makes β -actin at the furrow appear less intense than in uninduced cells with little to no beta actin at the cortex.

The quality of the blots in Fig S3 does not support the statement "expression of GFP fusion proteins at near endogenous levels (Supp Fig 3)." This expression at lower than wild type levels may have influenced the outcome of experiments where the wild type proteins were depleted. This potential artifact should be addressed in the text. Was expression similar in all the cells with dox? If not homogeneous, interpreting the data is complicated.

Response: We have supplied new blots (Supp Fig 3C, D) to demonstrate near endogenous levels of ectopic protein expression.

Fig. 5: the redistribution of the actin isoforms depending on the linker is impressive.

Page 11: Do you really mean “57.8±2.47% multi-nucleate cells in the GFP-DIAPH3 cell line compared to 4.82±0.840 for control siRNA treated cells?”

Response: The text has been amended to emphasize that the comparison is between GFP-DIAPH3 cells treated with either control or DIAPH3 siRNA.

Page 13: The small effect of depleting DIAPH3 on furrow ingression rate is surprising. Do the authors have any explanations? Divide the last paragraph into two parts.

Response: We agree. Based on currently favored models, one would expect that compromising β -actin at the furrow, where the cytokinetic ring is assembled, would have a more dramatic effect. However, as we (Chen *et al* 2021) and the Baum lab reported (Rodrigues *et al* 2015) and older models suggest (Wolpert 1960), polar relaxation has a role in cytokinesis and should be combined with equatorial contractility models. The simplest explanation of our observations is that perhaps polar relaxation, at least in the case of HeLa cells, has a greater contribution to furrow dynamics than was previously appreciated. We have commented to this effect in the revised discussion section of the manuscript.

Page 14, Fig 7A: It would be desirable to have better documentation (more examples, 3D reconstructions and perhaps colocalization with microtubules) to verify the important finding that “upon replacing beta-actin with gamma-actin at the furrow by expressing GFP-DIAPH3-1L in cells depleted of endogenous DIAPH3, phospho-myosin light chain was absent from the furrow membrane and instead concentrated at the spindle midzone.” In Fig 7B specify on the figure that the staining is for phosphorylated myosin-II. What mechanisms might explain the redistribution? Do beta- and gamma-actin have different affinities for myosin-II or any actin binding protein that might influence the localizations of myosin-II and RhoA?

Response: We have included further micrographs to demonstrate the relocalization of active nm-myosin-II to the spindle midzone (Fig. 7C). Based on the work of Manstein and co-workers any changes in affinity between actin isoforms and nm-myosin-II are small. However, those actin preparations were from baculovirus infected insect cells that are often reported to have up to 20% contamination by endogenous insect cell actins. With new strategies available for actin isoform preparation, future experiments should focus on repeating these experiments with the newer tools.

Page 16: Fig S5 was not included.

Response: We apologize for this omission and have now included it.

Page 17: The text states “there may be further complexity in the specificity process for some formins as in vitro the constitutively active DIAPH1 polymerized both beta- and gamma-actin homopolymeric filaments while in cellula full length DIAPH1 at the cell cortex produces only gamma-actin filaments.” The obvious difference is the presence of profilin in cells, which influences both nucleation and elongation by formins.

Response: We agree wholeheartedly, and this is a focus of ongoing experiments.

Minor concerns

Given the small numbers of cells, are 3 significant figures (for example “ingression index of 1.11 ± 0.0601 , 0.780 ± 0.0861 ” etc.) justified for most of measurements?

Response: We have now rounded up statistics to 2 decimal places.

Page 5 and elsewhere: Myosin is not a proper noun.

Response: We have made the corresponding changes to the text

Throughout: Is “in cellula” a real term?

Response: We use “in cellula” to differentiate between biochemical experiments (*in vitro*), tissue culture cell-based experiments (*in cellula*) and experiments in whole organisms (*in vivo*). Although some people have used “in cellulo” to describe events/experiments in tissue culture cells, “in cellulo” is not a latin term. Preferred phrases include “in cella” or more commonly “in cellula” (in a small cell), but there are active discussion groups online with respect to this. However, we would defer to those whose expertise in latin is greater. We have discussed this before with editors at Cell, JCB and JBC and they were OK with it, their priority was for neither “in vivo” nor “in situ” to be used to describe experiments in tissue culture cells. We would always defer to expert reviewers and journal editors and make changes accordingly.

Page 20, Table 3 and elsewhere. Paraformaldehyde is a solid which is converted formaldehyde for use as a fixative. Dilution factors are meaningless without knowing the stock concentration of the antibodies.

Response: We have corrected the methods and modified tables to include stock concentrations of antibodies.

Critique of the methods used to characterize actin polymerization:

Response: We thank Dr Pollard for these detailed and important comments and have modified the submission to remove instances of ‘nucleation’ throughout. We are acutely aware of the issues raised and addressing whether there is a difference between nucleation and elongation is extremely important. To achieve this, we need highly purified actin isoforms with as little as possible contamination of other isoforms. Recently, newer strategies have been developed to achieve this. Anecdotally we were warned of internal proteolysis sites which is a problem we encountered. We are currently trying to develop alternative streamlined approaches to generate the basic tools to rigorously perform the suggested experiments.

With respect to the filaments per field, we did indeed count filament ends. We have modified the methods to clarify our methodology.

The overall conclusions in Chen 2017 and 2021 and the current paper may be correct, but the methods used to characterize the formins make it difficult to interpret the data. In particular, none of the data actually show that DIAPH3 or DIAPH1 selectively stimulate nucleation of actin isoforms. The data do not rule out alternative hypotheses such as both formins nucleating equally well but then elongating filaments at much different rates. Therefore, using “assembly” rather than “nucleation” would safer.

Although doing proper experiments to characterize DIAPH1 and DIAPH3 is beyond the scope of this study, the authors should be aware of and acknowledge the following issues:

(1) Neither Chen paper (2017 and 2021) uses state-of-the-art quantitative assays for nucleation or elongation of actin filaments. During spontaneous polymerization assays such as Chen (2017) Figs 2 and 3b, both the rate of nucleation and the rate of elongation contribute to the accumulation of polymerized actin (measured by pyrenyl-actin fluorescence) over time. Elongation rates are measured most simply by TIRF microscopy. Knowing the elongation rate, one can calculate the concentration of growing filament ends from the slope at every point along the spontaneous polymerization time course. The change in the concentration of growing ends vs. time gives the nucleation rate. One can also estimate crudely the nucleation rate by observing the number of filaments by fluorescence microscopy (Chen 2017, Fig 5C). However, the samples in that experiment were taken after 20 min, very late in the polymerization process near steady state. Earlier time points would have been more informative. Furthermore, the method to measure “Filaments per field” is not explained. Is this filament ends or total polymer?

(2) Chen (2017) polymerized Ca-ATP-actin by adding KCl and MgCl₂, conditions where the slow exchange of Ca for Mg by actin monomers has been known to be rate limiting for nucleation for almost 40 years. To avoid this unphysiological divalent cation exchange reaction, virtually all spontaneous polymerization experiments are done by first exchanging Ca for Mg on

actin monomers and then adding KCl, EGTA and MgCl₂ to assemble Mg-ATP-actin monomers.

(3) Formins slow elongation, so one must know elongation rates to interpret the time course of spontaneous polymerization or measure nucleation rates (Chen 2017, Figs 2 and 3b). The data in those figures is certainly consistent with the CT-fragment stimulating nucleation but not definitive proof.

(4) All the polymerization assays are done without profilin, which is bound to actin monomers in cells. Profilin inhibits nucleation by formins more strongly than nucleation by pure actin monomers but increases the rate of elongation by formin constructs with FH1 domains such as the CT fragments in these papers. Therefore, the characterization of the two formins should have included polymerization with and without profilin.

Tom Pollard

REVIEWER COMMENTS

Reviewer #1 (Remarks to the Author):

The authors have successfully addressed our concerns.

Reviewer #3 (Remarks to the Author):

Shah/Wilde Nat Comm revised 3-24

The authors responded constructively to the first round of reviews and strengthened the revised paper. I still favor publication but am disappointed about a few points.

1. The authors responded to the questions about the Watanabe lab papers, but did not cite or discuss that highly relevant work.
2. The paper would be easier to read if the text were broken into shorter paragraphs with good topic sentences. Many paragraphs are still too long.
3. "Why does DIAPH3-1F assemble both beta- and gamma-actin on mitochondria, when previous work emphasized that DIAPH1 specifically assembles gamma-actin?" The authors clarified how the experiment was done, but do not address how truncated and full length constructs with the same FH2 domain could differ in this way. It makes me concerned about a misleading artifact of some sort.
4. "Fig 3: the FMNL3/actin crystal structure of Thompson et al. without contacts between the two actins is unlikely to be the form in solution." Unfortunately the authors retained this misleading ribbon diagram. Further, they add panel D with an alpha-fold2 model. This model cannot be better than the highly refined MD simulation model of Aydin et al. The fact that the linker clashes with actin in the MD model is due to its being disordered. A small additional simulation would eliminate this clash.
5. With respect to the blots in Fig S3 I asked "Was expression similar in all the cells with dox? If not homogeneous, interpreting the data is complicated." The authors did not respond about whether the expression was uniform from cell to cell.
6. On page 17, I noted "The obvious difference is the presence of profilin in cells, which influences both nucleation and elongation by formins." The authors agreed in their response but do not mention profilin in the revised text.
7. Use of "in cellula." Irrespective of any online discussion, why not use plain English such as "in live cells" or "in live cultured cells" rather than dreaming up new jargon?

Response to Reviewer 2.

1. The authors responded to the questions about the Watanabe lab papers, but did not cite or discuss that highly relevant work.

Response: We apologize for mis-understanding Dr Pollard. We interpreted his previous comment as a question directed towards us and not a request to address this in the manuscript. We have added the Watanabe reference to the revised manuscript that highlights that DIAPH3 *-/-* mice die *in utero* and DIAPH1 *-/-* mice reach adulthood. As to why mice expressing the truncations that lack the FH2 domain or either DIAPH1 or 3 reach adulthood, we do not know. No obvious molecular explanation is evident other than to suggest a simplistic model whereby the N-terminal half of the formins have other as yet unknown functions that can compensate during many stages of development. However, this is beyond the scope of our study.

2. The paper would be easier to read if the text were broken into shorter paragraphs with good topic sentences. Many paragraphs are still too long.

Response: We have made edits to the manuscripts. We note that Dr Pollard previously made the comment “The paper is clear and easy to read” and therefore see no need for major edits to jeopardize the manuscript’s flow. The differences in personal writing styles are to be expected and should be respected.

3. “Why does DIAPH3-1F assemble both beta- and gamma-actin on mitochondria, when previous work emphasized that DIAPH1 specifically assembles gamma-actin?” The authors clarified how the experiment was done, but do not address how truncated and full length constructs with the same FH2 domain could differ in this way. It makes me concerned about a misleading artifact of some sort.

Response: We discussed this in previous reviewer responses, especially in discussion with reviewer 1 in response to questions they asked. As stated before we don’t know why there is a difference with respect to the formin fragment targeted to the mitochondria and the full-length formin correctly targeted to the plasma membrane. We did outline potential reasons including localized regulation perhaps through post-translational modifications that are spatially restricted within the cell.

We stand by our statement that DIAPH1 is a γ -actin filament generator as this statement is based on the activity of the full-length endogenous DIAPH1 and the ectopically expressed GFP-DIAPH1 during cytokinesis.

Having said that, this is not the important point of the swap experiments which ask: “does the actin isoform specificity follow the FH2 domain and the linker”? We show that the answer is yes through 3 different approaches: an *in vitro* polymerization/pelleting assay, the mitochondrial targeting assay and most importantly in the context of the full-length protein during cytokinesis and find that the actin isoform specificity does follow the linker. In the context of the full-length protein targeted to the plasma membrane the activity again follows the linker, such that a DIAPH1 with a DIAPH3 linker now generates β -actin and not γ -actin.

Furthermore, a DIAPH3 with the DIAPH1 linker generates γ -actin and not β -actin. These observations in a dividing cell support the specificity model we propose and are not overly reliant on the mitochondrial targeting of actin isoform filaments using truncated formin constructs.

4. “Fig 3: the FMNL3/actin crystal structure of Thompson et al. without contacts between the two actins is unlikely to be the form in solution.” Unfortunately the authors retained this misleading ribbon diagram. Further, they add panel D with an alpha-fold2 model. This model cannot be better than the highly refined MD simulation model of Aydin et al. The fact that the linker clashes with actin in the MD model is due to its being disordered. A small additional simulation would eliminate this clash.

Response: We respectively disagree. Our use of these models needs to be taken in the context of the immediate prior results described in the manuscript and the desire to make our study accessible to newcomers to the field.

In the prior section of the manuscript we discovered that the FH2 domain possessed the actin isoform specificity. Our next step was to narrow down the region of the FH2 domain that was responsible for the actin isoform specificity. Rather than take a strategy of random mutagenesis or blindly mutating each sub-domain, we used existing information to develop a rational and targeted strategy to one sub-region of the FH2 domain. By using the Thompson structure, one of the few of a formin-actin co-crystal structure, we were able to compare the position of the whole of the FH2 domain with respect to an actin monomer. The crystal structure suggests that the actin N-terminus is most likely closer in space to the FH2 linker region than any other part of the FH2 domain. While the structure is not perfect and is missing some contacts, that study did identify critical FH2 domain residues required for the polymerization of actin filaments and therefore cannot be wholly dismissed as not reflecting at least some of the correct structures the complex will be capable of forming to perform its correct physiological activities. We also note that this is experimental data from an actual physical experiment and not data derived *in silico* from a computer simulation.

With respect to the data of Aydin *et al*, we do cite this study. However, we disagree with Dr Pollard that this model is better than the alpha fold or the crystal structure. As we pointed out before the Aydin model generates a non-physiological fold that could only be generated co-translationally and would prevent a dynamic interaction between the formins and the actin filament. We feel that this cannot be dismissed as a simple “clash”. The non-physiological fold (shown in the figure we provided before) will constrain the formin linker region directly adjacent to the residues that the Aydin model predicts form salt bridges with the actin N-terminus. This constraint in the model’s current form might be expected to significantly influence the interactions and the salt bridges. If this can be resolved with “small additional simulations” we would urge Dr Pollard to perform them.

Irrespective of our difference of opinion on the Aydin simulation data, our alpha fold model is important as it models DIAPH3 and actin, whereas the Aydin simulation modelled DIAPH1 and actin. The alpha fold model generates a complex that is not constrained by a non-physiological fold.

Importantly, when one takes all this information together, a consensus model forms: for multiple formins and using very varied experimental strategies, the FH2 linker sub-domain is the

part of the formin polypeptide chain closest to the variable actin N-termini and is therefore the most likely region of the formin to exert actin isoform specificity. By using and assessing these different models we generated a testable hypothesis that in subsequent sections of our manuscript we verify experimentally. Our intent in describing these models was to guide the reader through our process and rationale for focussing on the linker region vs other regions of the FH2.

5. With respect to the blots in Fig S3 I asked “Was expression similar in all the cells with dox? If not homogeneous, interpreting the data is complicated.” The authors did not respond about whether the expression was uniform from cell to cell.

Response: The short answer is yes: all cells individually expressed similar levels of GFP-formins upon Dox treatment, as judged by microscopy. We discussed this in response to reviewer 1’s questions and provided extra blots. Stable cell lines containing a single plasmid integration at a defined genomic site are generated through drug selection. The system works in such a way that only the stable integration of a single plasmid into the same position in the genome of all cells results in drug resistance. The consequences of any minor variations in expression that may occur will even out by quantifying large numbers of cells.

6. On page 17, I noted “The obvious difference is the presence of profilin in cells, which influences both nucleation and elongation by formins.” The authors agreed in their response but do not mention profilin in the revised text.

Response: Profilins are obvious potential candidates to influence a formin’s actin isoform specificity. However, and importantly, based on the co-crystal structure, profilin does not interact with the variable actin N-terminus, rather it interacts with a region of the actin monomer that is conserved across all actin isoforms. Based on this information and in the absence of experimental evidence of specificity in the interaction between different profilins and different actin isoforms, we do not feel justified in going out on a limb and proposing a highly speculative model that could be proved incorrect in the future. We prefer a more conservative approach such as not to contaminate the literature and be subsequently misquoted. Discussion of such speculative models would be best suited to a review where space would allow a fuller discussion of the pros and cons of this idea.

7. Use of “in cellula.” Irrespective of any online discussion, why not use plain English such as “in live cells” or “in live cultured cells” rather than dreaming up new jargon?

Response: *In cellula* is not jargon, rather a latin term, one much shorter than Dr Pollard’s proposed alternatives. A descriptive distinction is needed between living tissue culture cells and living cells in a whole organism, for example in a mouse. We prefer “*in cellula*” to make this distinction of experiments in tissue culture cells as opposed to living cells in a whole organism where the phrase “*in vivo*” is more apt. Fundamentally, journals should be the ones to determine their preferred usage.

REVIEWERS' COMMENTS

Reviewer #3 (Remarks to the Author):

As before, I recommend publication. The authors took some but not all of my suggestions that were meant to strengthen the paper.

1. Cite and comment on the prior work by the Watanabe lab. Done.
2. Break up long paragraphs. Declined. My comment that "the paper is clear and easy to read" is not inconsistent with making it even better by dividing long paragraphs on more than one topic, which the authors declined to do.
3. Isoform specificity of DIAPH1: I agree that it is acceptable not to understand the different results with two constructs.
4. Structure to introduce experiments showing that linker peptides contribute to isoform specificity: Thank you for pointing out the unnatural arrangement of the polypeptides in the Aydin MD simulation. I shared this problem with the other authors. Fortunately, an escape route was available and it did not interfere with the simulations, but it was not appropriate for your figure.
5. Homogeneous expression levels: reasonable responses.
6. The authors declined to mention profilin in the revised text. Not my recommendation but ok.
7. Use of "in cellula." The authors ask the journal to decide if this new terminology is better than plain English.